# IL-23 signaling prevents ferroptosis-driven renal immunopathology during candidiasis

Nicolas Millet [1,2], Norma V. Solis[1,2], Diane Aguilar[2], Michail S. Lionakis [3], Robert T. Wheeler [4], Nicholas Jendzjowsky [2,5] & Marc Swidergall [1,2,5] ✉

During infection the host relies on pattern-recognition receptors to sense invading fungal pathogens to launch immune defense mechanisms. While fungal recognition and immune effector responses are organ and cell type specific, during disseminated candidiasis myeloid cells exacerbate collateral tissue damage. The β-glucan receptor ephrin type-A 2 receptor (EphA2) is required to initiate mucosal inflammatory responses during oral *Candida* infection. Here we report that EphA2 promotes renal immunopathology during disseminated candidiasis. EphA2 deficiency leads to reduced renal inflammation and injury. Comprehensive analyses reveal that EphA2 restrains IL-23 secretion from and migration of dendritic cells. IL-23 signaling prevents ferroptotic host cell death during infection to limit inflammation and immunopathology. Further, host cell ferroptosis limits antifungal effector functions via releasing the lipid peroxidation product 4-hydroxynonenal to induce various forms of cell death. Thus, we identify ferroptotic cell death as a critical pathway of *Candida*-mediated renal immunopathology that opens a new avenue to tackle *Candida* infection and inflammation.

The first step in mounting an antifungal immune response is the recognition of extracellular pathogen-associated molecular patterns (PAMPs) of invading organism, such as *Candida albicans*, by soluble and membrane-bound pattern recognition receptors (PRRs)[1,2]. Following recognition, the effective control of *C. albicans* relies on several effector mechanisms and cell types to ensure fungal clearance[3–9]. Oral mucosal *Candida* infections (oropharyngeal candidiasis[10]) occur in over 750,000 individuals in the Unites States[11], while life-threatening bloodstream infections are estimated with 25,000 annual cases[12]. In contrast to mucosal candidiasis, in which IL-17-producing lymphocytes are crucial for host defense[13–17], effective immunity during disseminated candidiasis relies on myeloid phagocytes[2,18–20]. Although myeloid phagocytes are critical for host defense during disseminated candidiasis, their functions that are aimed to control fungal infections may also come at the cost of immunopathology[21,22]. In fact, excessive neutrophil accumulation in tissues late in the course of infection

is deleterious in mouse models of disseminated candidiasis[20,23]. Importantly, pathogenic immune cell effects can be observed in patients with invasive candidiasis with renal involvement, and in a subset of leukemia patients on corticosteroid therapy who develop a form of immune reconstitution inflammatory syndrome (IRIS) during the time of neutrophil recovery[24,25].

*C. albicans* is known to induce tissue injury and host cell death[20,21,26], e.g. in the kidney, a major target organ during disseminated candidiasis. Regulated host cell death (RCD) results in either lytic or non-lytic morphology, depending upon the signaling pathway[27]. Apoptosis is a non-lytic, and typically immunologically silent form of cell death[28]. On the other hand, lytic cell death is highly inflammatory[28–31], and includes necroptosis (alternative mode of RCD mimicking features of apoptosis and necrosis[32]), pyroptosis (RCD driven by inflammasome activation[27]), and ferroptosis (iron- and lipotoxicity-dependent form of RCD[30]). Inflammatory RCD depends on

[1]Division of Infectious Diseases, Harbor-UCLA Medical Center, Torrance, CA, USA. [2]The Lundquist Institute for Biomedical Innovation at Harbor-UCLA Medical Center, Torrance, CA, USA. [3]Fungal Pathogenesis Section, Laboratory of Clinical Immunology and Microbiology (LCIM), National Institute of Allergy and Infectious Diseases (NIAID), Bethesda, MD, USA. [4]Department of Molecular and Biomedical Sciences, University of Maine, Orono, ME, USA. [5]David Geffen School of Medicine at UCLA, Los Angeles, CA, USA. ✉e-mail: mswidergall@lundquist.org

the release of damage-associated molecular pattern (DAMPs) and inflammatory mediators[33]. RCD is increasingly understood to benefit the host[34], and *C. albicans* is known to induce inflammatory RCDs, such as necroptosis, and pyroptosis to promote inflammation[35,36]. Indeed, deficiencies in these pathways accelerate disease progression during fungal infection[36,37]. However, excessive inflammation results in renal immunopathology during candidiasis suggesting that other mechanisms or RCDs fine-tune immunopathology and fungal control. For instance, IL-23-mediated myeloid cell survival is crucial for maintaining abundant immune cells at the site of *Candida* infection to ensure efficient host protection[38]. Emerging data from various studies indicate an essential function of non-classical β-glucan recognition during fungal infections[39–44]. We recently found that EphA2 acts as a β-glucan receptor in the oral cavity that triggers the production of pro-inflammatory mediators via STAT3 and MAPK on oral epithelial cells, while EphA2 induces priming of neutrophil p47$^{phox}$ to increase intracellular reactive oxygen species (ROS) production to enhance killing of opsonized *C. albicans* yeast[39,40]. Although the function of this novel β-glucan receptor EphA2 is well established during oral mucosal *C. albicans* infection[39,40,45–47], the role of EphA2 during disseminated candidiasis is unknown. Here, we show that EphA2 promotes immunopathology during disseminated candidiasis. EphA2 deficient mice have reduced inflammation resulting in decreased renal injury and sepsis. EphA2 reduces migration of and IL-23 secretion from dendritic cells (DCs), while IL-23 receptor signaling inhibits host cell ferroptosis during candidiasis. Ferroptotic host cell death increases inflammation leading to immunopathology, and simultaneously limits antifungal effector functions within infected tissues.

## Results

### EphA2 deficiency increases tolerance during disseminated candidiasis

Being the core of the immune response, professional immune cells act as the most effective weapon to clear invading fungi. Dectin-1/CLEC7A is a major PRR of the C-type lectin family, predominantly expressed on myeloid-derived cells. Classical β-glucan recognition by Dectin-1 activates fungal phagocytosis and the production of pro-inflammatory cytokines[4,48]. Consistent with previous findings[49,50], Dectin-1 deficiency results in increased mortality in a mouse model of disseminated candidiasis (Fig. 1a). Although EphA2 recognizes β-glucan[39,44], and EphA2 deficiency results in increased susceptibility to oral fungal infection[39,40,46], *Epha2*$^{-/-}$ mice were more resistant during lethal *C. albicans* challenge (Fig. 1b, c). Since EphA2 is expressed on both stromal and hematopoietic cells[39,40,43,51,52], we generated bone marrow (BM) chimeric mice and determined their resistance to disseminated candidiasis (Fig. S1). Both, *Epha2*$^{+/+}$ mice reconstituted with *Epha2*$^{-/-}$ BM (knockout (KO)→wild-type (WT)) and *Epha2*$^{-/-}$ mice reconstituted with *Epha2*$^{+/+}$ BM (WT→KO) were more resistant during HDC compared to *Epha2*$^{+/+}$ mice reconstituted with *Epha2*$^{+/+}$ BM (WT→WT) (Fig. 1d). However, these chimeric mice were more susceptible than *Epha2*$^{-/-}$ mice reconstituted with *Epha2*$^{-/-}$ BM (KO→KO), which recapitulated the phenotype observed in global *Epha2*$^{-/-}$ mice (Fig. 1b), suggesting that EphA2 deficiency within cells of both, the hematopoietic and stromal compartments, is required for full protection against disseminated candidiasis.

Kidneys are a primary target organ of *C. albicans*, and invasion into the kidney medulla leads to loss of renal function and death[53,54]. Therefore, we determined the kidney fungal burden in WT and *Epha2*$^{-/-}$ mice 4 days post infection. Strikingly, no differences in renal fungal burden could be observed after 4 days of infection (Fig. 1e). We have previously shown that EphA2 activation triggers epidermal growth factor receptor-mediated invasion of oral epithelial cells[39,46]. The mouse model of disseminated candidiasis leads to rapid organ dissemination and clearance of >99% of the fungus from the bloodstream within the first hour after intravenous injection[55], while

*C. albicans* mutants defective in invasion have reduced kidney fungal burden[56,57]. To rule out a possible decrease of dissemination out of the bloodstream, we collected kidneys from WT and *Epha2*$^{-/-}$ mice 12 h post infection and enumerated fungal burden. Resistance and tolerance are two complementary host defense mechanisms that increase host fitness in response to invading *C. albicans*[58]. Since WT and *Epha2*$^{-/-}$ mice had similar renal fungal burden (Fig. 1f), we concluded that EphA2 deficiency enhances host tolerance, independent of fungal trafficking out of the bloodstream. Although EphA2 enhances neutrophilic killing of opsonized yeast[40], EphA2 has no effect on elimination of hyphae (Fig. S2), which is the dominant morphotype in kidneys after 12 h of systemic infection (Fig. S2).

Severe renal failure plays a major role in lethality of systemic *C. albicans* infection[9,59–61]. Therefore, we determined apoptotic areas in kidneys of WT and *Epha2*$^{-/-}$ mice infected *C. albicans* using terminal deoxynucleotidyl transferase dUTP nick end labeling (TUNEL) staining[54]. Although apoptosis could be detected in kidneys of both mouse strains, the overall apoptotic areas decreased in infected *Epha2*$^{-/-}$ mice (Fig. 1g and Fig. S3). Next, we assessed serum neutrophil gelatinase-associated lipocalin (NGAL), a marker of acute kidney injury[54,62], as well as NGAL levels in infected kidneys. NGAL was strongly present in WT mice, while this kidney injury marker was reduced in *Epha2*$^{-/-}$ mice (Fig. 1h). During disseminated candidiasis, both systemic inflammation as well as rapid deterioration of the infected host resembles hyper-inflammatory sepsis[63]. Therefore, we measured the sepsis marker soluble triggering receptor expressed on myeloid cells (TREM1)[64] in serum of infected mice. TREM1 was significantly reduced in *Epha2*$^{-/-}$ mice compared to WT mice (Fig. 1i). Given that EphA2 promotes inflammation during oral *C. albicans* infection[39,40,45–47], we assessed the contribution of EphA2 in a mouse model of zymosan (β-glucan) - induced acute kidney injury (AKI), in which immune cells and inflammation exert essential roles in kidney damage[65,66]. We found that EphA2 deficient mice were more resistant to AKI (Fig. 1j). Together, this data suggest that EphA2 promotes inflammation to accelerate disease progression during disseminated candidiasis.

### EphA2 promotes renal inflammation during disseminated candidiasis

Although the host immune response is required to control *C. albicans* infections, the inflammatory response causes significant collateral tissue damage. Therefore, we determined the cytokine and chemokine response in infected kidneys in WT and *Epha2*$^{-/-}$ mice. *Epha2*$^{-/-}$ mice had reduced kidney levels of several pro-inflammatory cytokines, including TNFα, IL-1β, and IL-6 (Fig. 2a). By contrast, we found that *Epha2*$^{-/-}$ mice had increased levels of IL-23, IFNγ, and IL-4 (Fig. 2b). Next, we assessed renal leukocyte infiltration during infection. *Epha2*$^{-/-}$ mice had decreased accumulation of neutrophils and monocytes (Fig. 2c and Fig. S4). Although no differences in total macrophage numbers were observed (Fig. 2c) the live macrophage population (% of CD11b$^+$) increased in *Epha2*$^{-/-}$ mice compared to WT (Fig. 2d). Consistent with previous observations[67], EphA2 deficiency increased the accumulation of dendritic cells (DCs) during infection (Fig. 2e–g). These experiments suggested that EphA2 is critical to promote renal inflammation during disseminated candidiasis.

### EphA2 deficiency reduces renal ferroptosis during disseminated candidiasis

To comprehensively evaluate the transcriptional response of *Epha2*$^{-/-}$ mice during *C. albicans* infection, we performed RNA sequencing of infected kidneys. Consistent with our findings that EphA2 deficient mice had reduced renal apoptosis (Fig. 1g) and reduced inflammation (Fig. 2), KEGG pathway analysis revealed downregulation of genes involved in apoptosis and several immune response pathways (Fig. 3a). Using Gene Set Enrichment Analysis (GSEA) we found that genes

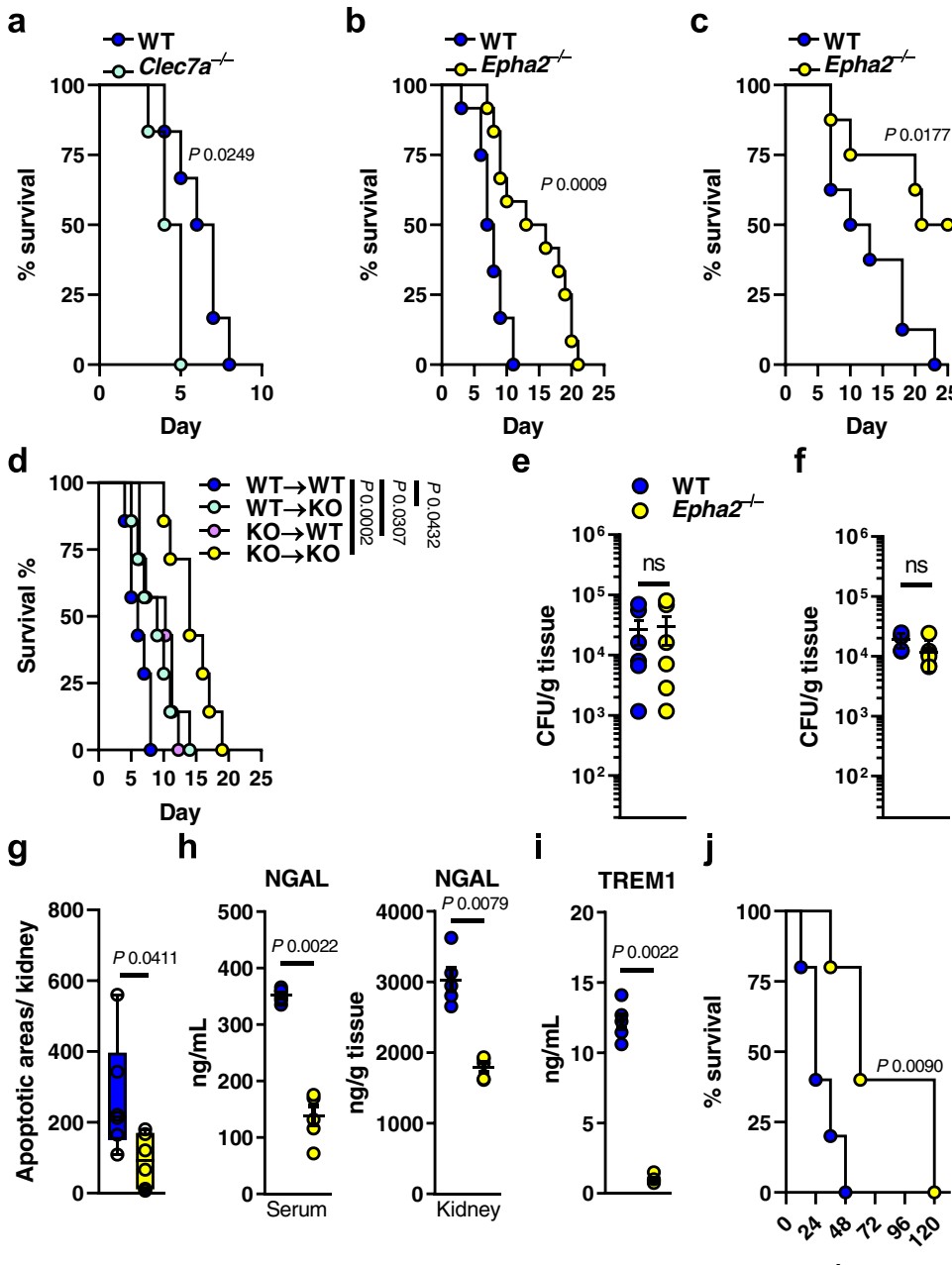

**Fig. 1 | EphA2 promotes disease progression during disseminated candidiasis.**
**a** Survival of wild-type and *Dectin-1*[-/-] *(Clec7a*[-/-]*)* mice infected intravenously with $2.5 \times 10^5$ SC5314 *C. albicans*. *N* = 6; combined data of two independent experiments.
**b**, **c** Survival of wild-type and *Epha2*[-/-] mice infected intravenously with $2.5 \times 10^5$ (**b**; *N* = 12; combined data of two independent experiments) or $1.25 \times 10^5$ (**c**; *N* = 8; combined data of two independent experiments) SC5314 *C. albicans*. **d** Survival of bone marrow chimeric mice following infection with $2.5 \times 10^5$ SC5314 *C. albicans*. *N* = 7; combined data of two independent experiments. *Epha2*[+/+] mice reconstituted with *Epha2*[+/+] BM (WT→WT), *Epha2*[+/+] mice reconstituted with *Epha2*[-/-] BM (KO→WT), *Epha2*[-/-] mice reconstituted with *Epha2*[+/+] BM (WT→KO), and *Epha2*[-/-] BM to and *Epha2*[-/-] mice (KO→KO). Mantel–Cox Log-Rank test. Kidney fungal burden of

infected mice after **e** 4 days and **f** 12 h of infection with $2.5 \times 10^5$. Results are median (*N* = 6) ±SEM of two independent experiments combined. ns, no significance. Two-tailed Mann–Whitney Test. **g** Apoptotic areas per kidney after 4 days of infection determined by TUNEL staining. *N* = 6; combined data of two independent experiments. Two-tailed Mann–Whitney Test. Box-and-whisker plots indicating median, 25th/75th percentiles, and the minimum/maximum values. **h** Serum and kidney NGAL, and **I** serum TREM1 levels after 3 days of infection. Results are median (*N* = 6) ±SEM of two independent experiments combined. Two-tailed Mann–Whitney Test. **j** Survival of wild-type and *Epha2*[-/-] mice injected intraperitoneally with 750 mg/kg of zymosan. *N* = 5; combined data of two independent experiments. Mantel–Cox Log-Rank test.

involved in ferroptosis were significantly enriched in infected kidneys from WT mice compared to *Epha2*[-/-] mice (Fig. 3b). In tumor cells, SLC7A11-mediated cystine uptake promotes GPX4 protein synthesis to reduce sensitivity to ferroptotic cell death[68,69]. Therefore, we determined the two anti-ferroptotic genes *SLC7a11* and *GPX4*, as well as *LYZ2*, a myeloid cell lineage marker, using RNAscope (Fig. 3c). *SLC7a11* and *GPX4* RNAscope particles were reduced in infected kidneys from

*Epha2*[-/-] mice (Fig. 3d). Furthermore, *SLC7a11* and *GPX4* particles were enriched in myeloid cells (*LYZ2*[+]) in kidney sections from WT compared *Epha2*[-/-] mice (Fig. 3e, f). Using immunofluorescence, we confirmed that GPX4 protein expression was reduced in *Epha2*[-/-] mice in infected tissue (Fig. 3g and Fig. S5). Ferroptosis is a lipid peroxidation-driven form of RCD[70]. Therefore, we determined the level of lipid peroxidation using 4HNE staining. While kidney sections of WT mice showed

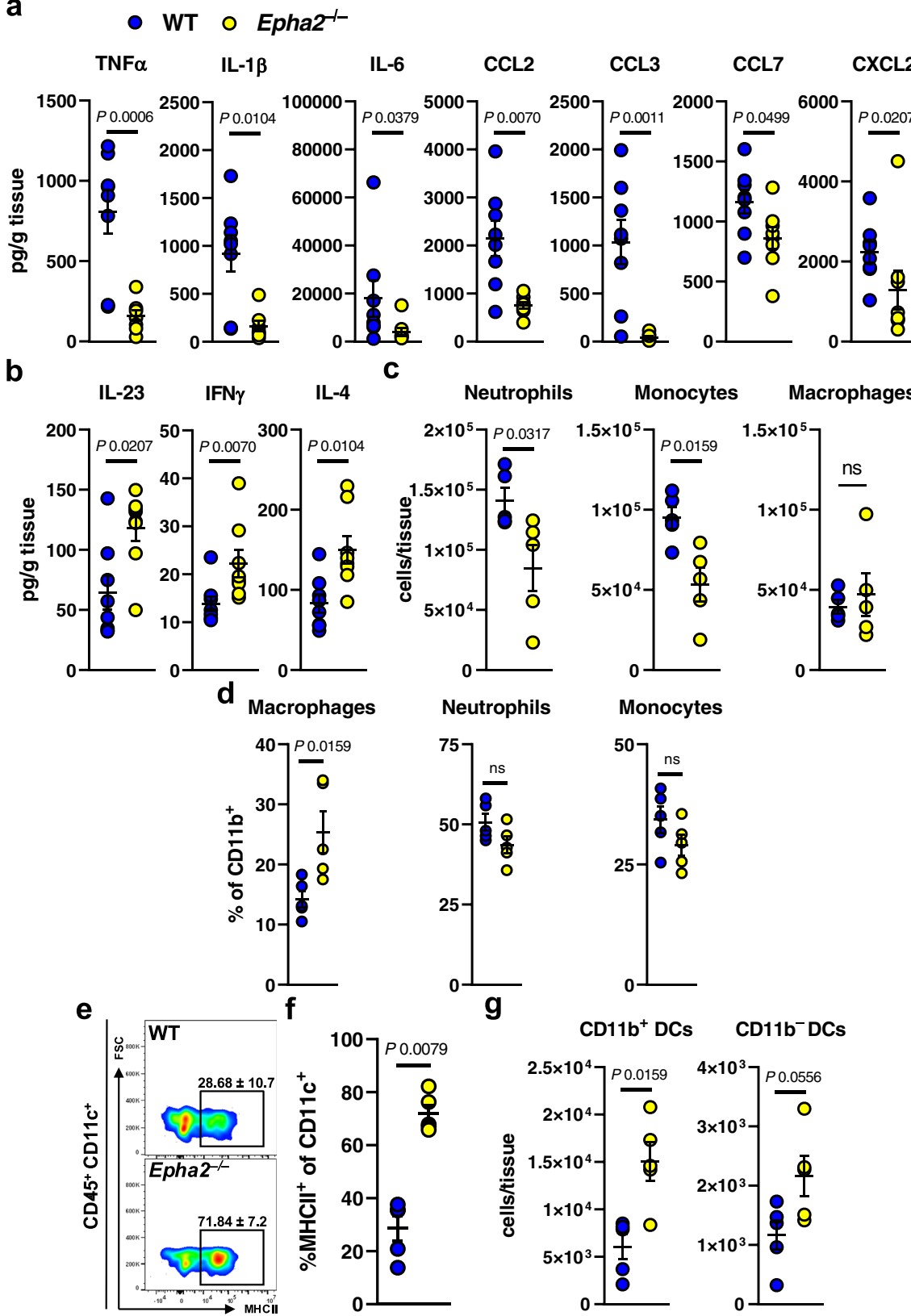

strong lipid peroxidation, sections of *Epha2⁻/⁻* mice had low levels of ferroptosis (Fig. 3h and Fig. S5) suggesting that during candidiasis host cells undergo excessive lipid peroxidation although anti-ferroptotic mechanisms are upregulated in myeloid cells. Since *LYZ⁻* cells in kidney sections from WT and *Epha2⁻/⁻* mice had comparable *GPX4* expression, we tested whether EphA2 deficiency in macrophages

(*LYZ2⁺*) results in decreased ferroptosis. Using the lipid peroxidation sensor BODIPY, as well as 4HNE staining, we showed that bone marrow-derived macrophages (BMDMs) from *Epha2⁻/⁻* mice underwent similar magnitudes of ferroptosis compared to WT BMDMs (Fig. S6). Collectively, this data suggest that EphA2 promotes ferroptotic cell death via an extrinsic pathway during candidiasis.

**Fig. 2 | EphA2 deficiency reduces renal neutrophil and monocyte recruitment, but increases DC accumulation during candidiasis. a**, **b** Level of indicated cytokines in kidneys after 3 days of infection. $N = 8$; combined data of two independent experiments. Two-tailed Mann–Whitney Test. **c** Accumulation of neutrophils, monocytes, and macrophages (total numbers) in the kidney of wild type and $Epha2^{-/-}$ mice after 3 days of infection. $N = 5$, combined data of two independent experiments. ns; No Significance. Two-tailed Mann–Whitney Test. **d** Percent of macrophages, neutrophils, and monocytes of CD11b$^+$ cells. $N = 5$, combined data of

two independent experiments. ns, no significance. Two-tailed Mann–Whitney Test. **e**, **f** Representative flow cytometry plots of MHCII expression and frequencies of MHCII-expressing cells in infected kidneys after 3 days of infection. $N = 5$; combined data of two independent experiments. Two-tailed Mann–Whitney Test **g** Accumulation of CD11b$^+$ and CD11b$^-$ dendritic cells in kidneys of wild type and $Epha2^{-/-}$ mice after 3 days of infection ($N = 5$). Two-tailed Mann–Whitney Test. Results are median ± SEM.

## Ferroptotic cell death exerts inflammation and promotes disease severity during candidiasis

To investigate whether ferroptotic cell death exacerbates inflammation and disease progression during fungal infection, we first determined that *C. albicans* induces ferroptosis in BMDMs and renal tubular epithelial cells (RTECs) (Fig. S7). Inhibition of ferroptosis using the selective inhibitor Ferrostatin-1 (Fer-1)[71] increased survival of BMDMs and RTECs during *C. albicans* interactions (Fig. 4a, b and Fig. S8). This finding is consistent with previous reports showing that Fer-1 treatment reduces macrophage cell death during *Histoplasma capsulatum* infection[72]. Increased BMDM survival was associated with increased *C. albicans* killing (Fig. 4c). Furthermore, inhibition of ferroptosis reduced cytokine secretion in BMDMs (Fig. 4d) and RTECs (Fig. 4e) suggesting that ferroptosis exerts inflammation during *C. albicans* infection. Next, we tested whether ferroptosis in uninfected macrophages promotes inflammation using RSL3, a GPX4 inhibitor and potent ferroptosis inducer. While RSL3 induced BMDM cell death (Fig. 4f), inflammation, measured by TNFα release, depended on fungal infection (Fig. 4g). Thus, RSL3 provokes a type of ferroptosis that does not include inflammatory cytokine production, whereas inflammation exclusively occurred during *Candida*-stimulated ferroptotic macrophage cell death. To analyze the contribution of ferroptotic cell death to the pathogenicity of disseminated candidiasis, we infected WT mice with *C. albicans* followed by daily treatment with Fer-1. Mice treated with Fer-1 were less susceptible to *C. albicans* challenge compared to vehicle control mice (Fig. 4h, i). Consistent with the in vitro data (Fig. 4a–e), Fer-1 treatment decreased renal fungal burden (Fig. 4j), reduced serum NGAL and TREM1 (Fig. 4k, l), and inflammatory cytokines TNFα, IL-1β, and IL-6 (Fig. 4m). Collectively, disseminated candidiasis induces ferroptosis in various host cell types to promote inflammation and disease progression.

## EphA2 and JAK signaling limit IL-23 secretion in DCs

Following systemic *C. albicans* infection, DCs produce IL-23 to stimulate natural killer (NK) cell activity[8]. Accordingly, renal IL-23 levels correlated with increased CD11b$^+$ DC infiltration during infection in $Epha2^{-/-}$ mice (Fig. 5a). The kidney hosts various DC subsets[73], and some of these subsets produce IL-23, including tissue-migratory CD103$^-$ CD11b$^+$ and CD103$^+$ CD11b$^-$ DCs[74]. In consequence, we found an increase in CD103$^-$ CD11b$^+$ DCs in infected kidneys of $Epha2^{-/-}$ compared to WT mice, while no differences where observed in CD103$^+$ CD11b$^-$ DCs, as well as plasmacytoid DCs (pDCs) populations (Fig. 5b and Fig. S9), which are required for antifungal defense[75–77]. DCs derived from $Epha2^{-/-}$ BM secreted more IL-23 when stimulated with β-glucan (Fig. 5c), while TNFα levels were unaffected (Fig. S10). Next, we examined the transcriptional response of β-glucan stimulated DCs using RNA sequencing. KEGG pathway mapping revealed downregulation of genes associated with Janus-associated kinase (JAK)- Signal transducers and activators of transcription (STAT) signaling in $Epha2^{-/-}$ BMDCs, while genes of the peroxisome proliferator-activated receptors (PPAR) pathway were upregulated (Fig. 5d). To investigate the contribution of JAK-STAT and PPAR signaling to IL-23 secretion in DCs, we treated WT BMDCs with Ruxolitinib (JAK1/2 inhibitor), the PPARγ antagonist GW9662, and the PPARγ agonist Rosiglitazone followed by β-glucan stimulation. While

PPARγ stimulation or inhibition had no effect on IL-23 secretion, Ruxolitinib increased IL-23 levels in supernatants during β-glucan stimulation (Fig. 5e) suggesting that β-glucan-induced JAK-STAT signaling reduces IL-23 secretion in DCs.

## IL-23 signaling inhibits ferroptosis during disseminated candidiasis

Besides stimulating NK cells[8], IL-23 secures myeloid cell survival during candidiasis[38]. It is thought that this mechanism is key for maintaining sufficient numbers of phagocytes at the site of infection to ensure efficient host protection[38]. The link between myeloid cell survival and IL-23 was intriguing since IL-23 receptor downstream targets have been associated to counteract ferroptotic cell death[78,79]. The IL-23 receptor is expressed by several immune cells[38], including CD11b$^+$ F4/80$^+$ macrophages (Fig. S11), during disseminated candidiasis. To test if IL-23 prevents macrophage ferroptosis during *C. albicans* infection, we measured total cell fluorescence of oxidized C11-BODIPY in infected macrophages. Exogenous IL-23 reduced macrophage lipid peroxidation during *C. albicans* interaction (Fig. 6a). Furthermore, treatment with IL-23 increased macrophage survival (Fig. 6b), their *C. albicans* killing capacity (Fig. 6c), and reduced inflammation during infection (Fig. 6d), as seen in macrophages treated with the ferroptosis inhibitor (Fig. 4). This data suggested that IL-23 signaling reduces ferroptotic macrophage cell death during *C. albicans* infection. RTECs respond to IL-23 by increasing IL-23R expression followed by signaling to promote local kidney inflammation[80]. Given that IL-23 levels (IL23 p19) are abundant in kidneys during disseminated candidiasis (Fig. 6e) we tested whether IL-23 signaling will reduce ferroptotic cell death in RTECs as seen in macrophages. As described by the authors in ref. 80, IL-23 treatment increased IL-23R expression in RTECs (Fig. S12). Incubation of RTECs with recombinant human IL-23 decreased *Candida*-mediated cell death in a time dependent manner (Fig. 6f). Next, we assessed whether IL-23 signaling reduces ferroptosis in RTECs, as seen in macrophages during infection. Prolonged exposure of RTECs to IL-23 decreased *Candida*-induced lipid peroxidation (Fig. 6g, h). Furthermore, IL-23 treatment reduced cytokine secretion of RTECs during *Candida* infection (Fig. 6i). To investigate anti-ferroptotic IL-23 signaling during candidiasis in vivo, we systemically infected $Il23r^{WT/WT}$ and $Il23r^{GFP/GFP}$ mice, which are unresponsive to IL-23[81]. $Il23r^{GFP/GFP}$ mice had increased renal fungal burden (Fig. 7a), as well as excessive ferroptosis indicated by 4HNE staining (Fig. 7b, c and Fig. S13). Consequently, IL-23R deficiency increased kidney injury and sepsis (Fig. 7d, e), while exaggerated inflammation (Fig. 7f and Fig. S14). In parallel, we treated *Candida*-infected WT mice with recombinant murine IL-23 for 3 consecutive days (Fig. 8a). IL-23 treatment increased the median survival by >65% (6 vs. 10 days) (Fig. 8a) and reduced the renal fungal burden (Fig. 8b). Next, we determined ferroptosis in infected kidneys by staining section for 4HNE. Treatment with rmIL-23 decreased lipid peroxidation compared to vehicle control (Fig. 8c, d and Fig. S13). Moreover, mice treated with IL-23 had reduced kidney injury (Fig. 8e), sepsis (Fig. 8f), and inflammation (Fig. 8g). IL-23 promotes the development of an IL-17−producing CD4$^+$ T cell subset (Th17)[82]. However, no differences in Th17 surrogate markers IL-17A and IL-22 were observed (Fig. S15) suggesting that reduction in host ferroptosis, inflammation, and

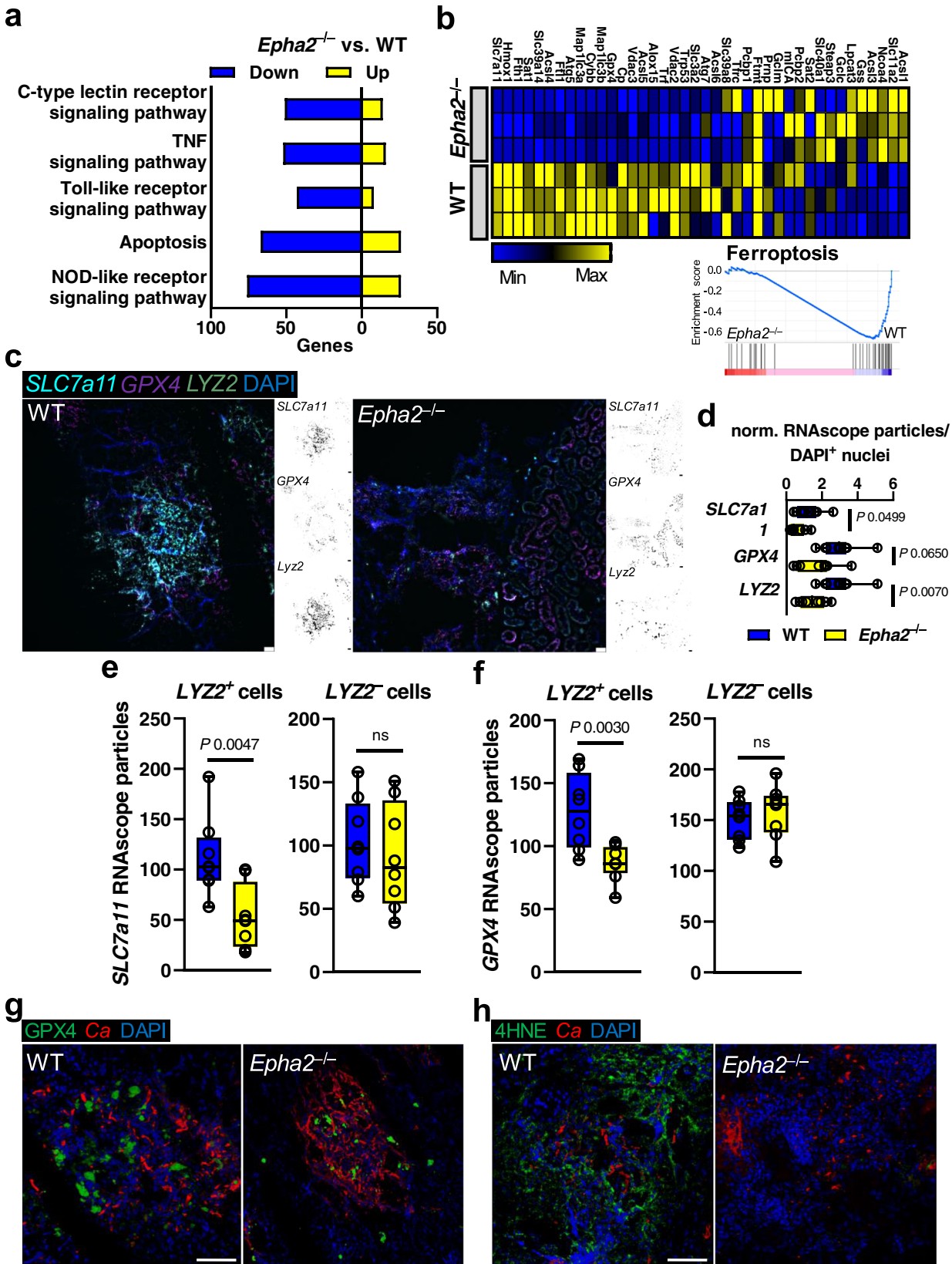

renal fungal burden is independent of the IL-23/Th17 axis. To determine if increased IL-23 levels contribute to tolerance of *Epha2*<sup>–/–</sup> mice during candidiasis we depleted IL-23 p19 during infection. IL23 p19 depletion decreased survival of *Epha2*<sup>–/–</sup> mice during candidiasis (Fig. 9a), increased renal fungal burden (Fig. 9b), and exaggerated renal ferroptosis (Fig. 9c, d and Fig. S13). Furthermore, IL-23 p19

depletion in *Epha2*<sup>–/–</sup> mice increased kidney injury (Fig. 9e), while TREM1 levels were similar after 3 days of infection (Fig. 9f). Further, depletion of IL-23 p19 increased inflammation indicated by TNFα and IL-6 (Fig. 9g). Collectively, we show that IL-23 signaling prevents inflammatory ferroptosis in host cells to improve disease outcomes during disseminated candidiasis.

**Fig. 3 | EphA2 promotes ferroptotic host cell death during candidiasis. a** Up- and down-regulated number of genes of KEGG pathways. RNASeq was performed on mRNA isolated from kidneys of WT and *Epha2*⁻/⁻ mice after 3 days of infection. *N* = 3 per mouse strain. **b** GSEA of ferroptosis pathway genes. Heatmap of ferroptosis gene expression analysis in WT and *Epha2*⁻/⁻ mice. Normalized enrichment score is shown on Y-axis. **c** Representative RNAscope image of *SCL7a11*, *GPX4*, and *LYZ2* expression in infected kidneys after 3 days of infection. Scale bar 50 μm. *N* = 8; 4 independent kidney sections of two animals per group. **d–f** Quantification of RNAsope particles. **d** Normalized particles of DAPI⁺ nuclei. *SLC7a11*⁺ particles of *LYZ2* positive and negative cells. *N* = 8; 4 independent kidney sections of two animals per group. Box-and-whisker plots indicating median, 25th/75th percentiles,

and the minimum/maximum values. Two-tailed Mann–Whitney Test. **e** *SLC7a11*⁺ and **f** *GPX4*⁺ particles of *LYZ2*-positive and negative cells. *N* = 8; 4 independent kidney sections of two animals per group. Box-and-whisker plots indicating median, 25th/75th percentiles, and the minimum/maximum values. Two-tailed Mann–Whitney Test. **g** GPX4 protein expression in infected kidneys after 3 days of infection. GPX4 shown in green, *C. albicans* (*Ca*) in red. Tissue is visualized using DAPI. Scale bar 100 μm. *N* = 7; independent animals. **h** Lipid peroxidation in infected kidneys after 3 days of infection using 4HNE. 4HNE shown in green, *C. albicans* (*Ca*) in red. Tissue is visualized using DAPI. Scale bar 100 μm. *N* = 7; independent animals.

## Ferroptosis product 4HNE induces various types of host cell death

The lipid peroxidation product 4HNE has been implicated in the etiology of pathological changes as a key mediator of oxidative stress-induced cell death. This reactive aldehyde species is a main product of lipid peroxidation during ferroptosis. We hypothesized that the release of 4HNE during immune cell ferroptosis will impair antifungal effector functions in adjacent cells. Thus, we tested whether macrophages release 4HNE during *C. albicans* infection. 4HNE release was observed during *C. albicans*-macrophage interactions, which was blocked by IL-23-mediated ferroptosis inhibition (Fig. 10a). Exogenous 4HNE inhibited fungal killing in a dose dependent manner (Fig. 10b) suggesting that ferroptotic cell death and consequently 4HNE release will diminish antifungal effector functions within an infected tissue. Of note, exogenous 4HNE had minimal effect on fungal growth at used concentration (Fig. S16). Exposure of uninfected macrophages to 4HNE induced ferroptosis (Fig. 10c, d), macrophage blebbing (Fig. 10c, e), a hallmark of apoptosis, as well as necroptosis indicated by MLKL activation (Fig. 10f and Fig. S17). Furthermore, infection with *Candida* yeast followed by 4HNE exposure accelerated pyroptosis induction indicated by gasdermin-D cleavage (Fig. 10g and Fig. S17). Consistent with these findings *Epha2*⁻/⁻ mice had reduced expression of key genes in theses cell death pathways (Fig. S18). Collectively, host cell ferroptosis not only impairs fungal clearance intrinsically, but also indirectly by releasing lipid peroxidation products into the infected environment; thus, promoting cell death and reduce antifungal effector functions.

## Discussion

Distinct RCD mechanisms promote the resolution of infection by destroying intracellular niches which benefit the pathogen, and to coordinate an appropriate innate immune response thereafter[28]. However, the RCD ferroptosis has been implicated in the development of many diseases[83–85]. Although inflammation is required to fight infections, a reduction in myeloid cell-mediated immunopathology may lead to pathogen tolerance, a phenomenon whereby the host is able to better resist infection by reducing immune cell-mediated tissue damage[21,86]. Our findings uncover a mechanism linking ferroptosis to immunopathology during candidiasis (Fig. S19). Here we show that myeloid and stromal cells undergo ferroptotic RCD to accelerate inflammation during fungal encounter. Macrophage and RTEC ferroptosis is limited by IL-23 receptor signaling; thus safeguarding efficient fungal clearance and controlled inflammation.

In response to reduced inflammatory signals, classical macrophages (M1 state) are more resistant to pro-ferroptotic treatment with a specific GPX4 inhibitor[87]. Our data suggest that ferroptotic cell death induced by enzyme inhibitors differs from pathogen-induced ferroptotic myeloid cell death in their inflammatory capacity. Indeed, sensitization to ferroptotic cell death is regulated by various mechanical stimuli[88]. Protrusive force is generated during phagocytosis[89], and immune cells respond to mechanical forces during the polarized growth of fungal hyphae[90]. Likewise, fungal β-glucan recognition induces mTOR signaling in monocytes[91], which has been associated with increased sensitivity to ferroptosis[92]. Hence, mechanical forces

during infection and consequently activation of distinct downstream signaling pathways sensitize myeloid cells to undergo inflammatory ferroptotic cell death. Further, iatrogenic immunosuppression, such as corticosteroid therapy, a risk factor for invasive candidiasis[23,93], sensitize host cells to ferroptosis[94]. Whether steroid-mediated sensitization facilitates local ferroptosis during *Candida*–host cell interactions to accelerate fungal dissemination needs to be investigated.

While the production of reactive oxygen species (ROS) is a key aspect of phagocyte-mediated host responses during *C. albicans* infection, an increase in ROS may cause lipid peroxidation and ferroptosis[95]. Here, we show that ferroptotic cell death benefits fungi by reducing the killing capacity of macrophages. Lipid peroxidation during ferroptosis results in production of 4HNE, which is able to react with primary amines on proteins or DNA to form crosslinks[96]. Exogenous 4HNE impairs the PKC signaling pathway in RAW macrophages[97], a critical mediator of antifungal host defense[98]. Macrophage-*Candida* interactions lead to 4HNE release, while exogenous 4HNE triggers several cell death mechanisms in macrophages, including ferroptosis, pyroptosis, necroptosis, and apoptosis. At the same time 4HNE inhibits macrophage-mediated fungal killing; thus, limiting antifungal effector functions within infected tissues. Ferroptotic cancer cells fail to elicit immune protection despite the release of DAMPs and cytokines[99] suggesting that lipid peroxidation products released by and/or at the surface of ferroptotic cancer cells may interfere with cancer immunotherapies.

IL-23 has received significant interest as a therapeutic target for a number of autoimmune conditions in recent clinical trials[100]. However, IL-23 has established roles during antifungal immunity[101]. IL-23 expression is strongly induced in response to *C. albicans* via the C-type lectin and TLR pathways[102] and is best known for regulating IL-17 production by T cells and innate lymphoid cells at epithelial barriers[100]. IL-23 binds to IL-12Rβ1 and IL-23R followed by receptor complex signaling via JAK2 and Tyk2[101,103,104]. In a mouse model of disseminated candidiasis, pre-treatment with tofacitinib[105] and ruxolitinib[106] (JAK inhibitors) increase susceptibility and fungal burden, while inflammation increases when therapy is started at the onset of disease[106]. Furthermore, the IL-23 signaling cascade activates several STAT members, including STAT3[107,108]. In tumor cells, STAT3 inhibition induces ferroptosis via Nrf2-GPX4 signaling[78], while STAT3 activation suppresses expression of ACSL4, an enzyme that enriches membranes with long polyunsaturated fatty acids and is required for ferroptosis[79]. Nur et al. showed that IL-23 secures survival of myeloid cells during candidiasis by inhibiting apoptosis[38]. Here, we demonstrate a non-canonical role of IL-23 signaling in inhibiting macrophage and RTEC ferroptosis during fungal infection. Historically, cell death pathways have long been considered to function in parallel with little or no overlap. However, it is currently known that lytic and non-lytic RCDs, such as apoptosis, necroptosis, pyroptosis, and ferroptosis are tightly connected, and can cross-regulate each other[27]. For instance the anti-ferroptotic enzyme GPX4 reduces lipid peroxidation-dependent caspase-11 activation and gasdermin D-mediated pyroptosis during polymicrobial infection[109]. Further, STAT3 activation (downstream of IL-23R) limits ferroptosis, pyroptosis, and necroptosis[78,110,111]. This suggest

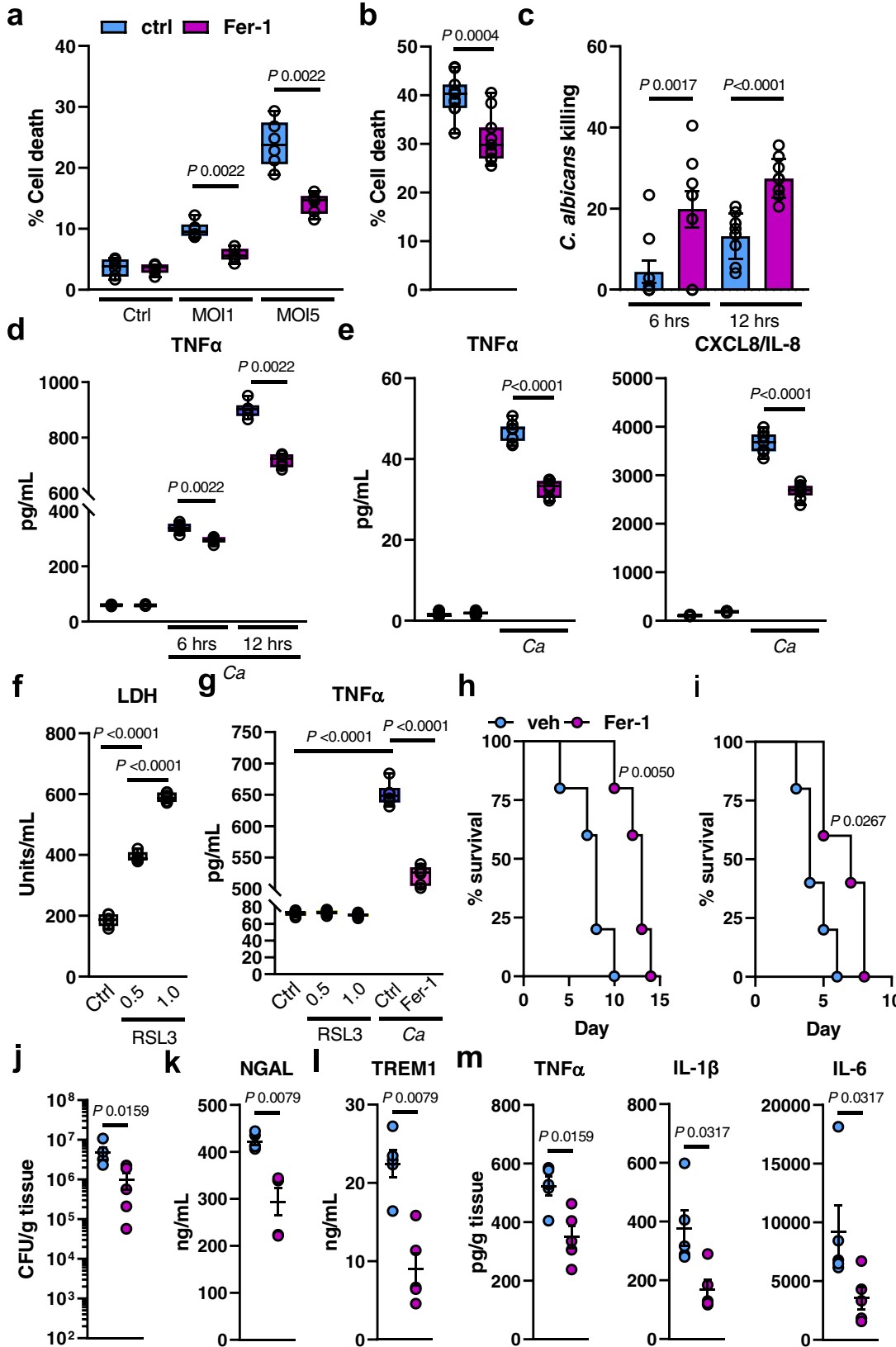

that different types of cell death depend on the stimulant, either infectious agent or drug, the cell type, and the environment, but share similar downstream signals and molecular regulators.

While IL-23 inhibits ferroptotic host cell death, some cytokines, such as IFNγ, drive ferroptosis via STAT1 signaling in tumor cells[112]. Whether the inflammatory tissue environment during fungal infection

and their corresponding cytokines and chemokines sensitize immune and stromal cells to ferroptotic cell death needs to be determined.

During acute mucosal *Candida* infection, epithelial EphA2 activates distinct chemokines, cytokines, and host defense peptides to prevent fungal invasion[39,45,46], while EphA2 in neutrophils is required for p47[phox] priming to enhance fungal clearance[40]; thus, EphA2 in

**Fig. 4 | Ferroptotic cell death exerts inflammation, decreases macrophage-mediated fungal killing, and promotes disease progression during candidiasis.** **a** BM-derived macrophages were treated with 10 µM Fer-1 for 1 h followed by interaction with *C. albicans* (MOI 1 and 5) for 4 h. PI$^+$ (dead) cells were determined by gating on F4/80$^+$ cells. $N=3$; duplicate. Two-tailed Mann–Whitney Test. Box-and-whisker plots indicating median, 25th/75th percentiles, and the minimum/maximum values. **b** Renal tubular epithelial damage determined by specific $^{51}$Cr release. $N=3$; triplicate. Two-tailed Mann–Whitney Test Box-and-whisker plots indicating median, 25th/75th percentiles, and the minimum/maximum values. **c** *C. albicans* killing of macrophages treated with Fer-1. MOI 1. $N=3$; triplicate. Two-tailed Mann–Whitney Test. Bar graphs show mean values ± SEM. **d** TNFα secretion of BM-derived macrophages during *C. albicans* infection (MOI 1) for the indicated time points. $N=3$, duplicate. Two-tailed Mann–Whitney Test. Box-and-whisker plots indicating median, 25th/75th percentiles, and the minimum/maximum values. **e** CXCL8 and TNFα secretion of RTECs in the presence and absence of Fer-1 during *C. albicans* infection. MOI 5. Two-tailed Mann–Whitney Test. Box-and-whisker plots indicating median, 25th/75th percentiles, and the minimum/maximum values.

**f** Lactate dehydrogenase (LDH) release of RSL3 (0.5 and 1 µM) treated BM-macrophages. Mann–Whitney Test. $N=3$ in duplicate. Box-and-whisker plots indicating median, 25th/75th percentiles, and the minimum/maximum values. **g** TNFα release of RSL3-treated or *C. albicans*-infected macrophages (MOI 1) in the presence or absence of Fer-1 (10 µM). 12 h. Two-tailed Mann–Whitney Test. $N=3$; duplicate. Box-and-whisker plots indicating median, 25th/75th percentiles, and the minimum/maximum values. **h** Survival of WT mice treated daily with 10 mg/kg Fer-1 or vehicle control. Inoculum $1.25 \times 10^5$ *C. albicans*. $N=5$; two independent experiments per inoculum. Mantel–Cox Log-Rank test. **i** Survival of WT mice treated daily with 10 mg/kg Fer-1 or vehicle control. Inoculum $2.5 \times 10^5$ *C. albicans*. $N=5$; two independent experiments per inoculum. Mantel–Cox Log-Rank test. **j** Renal fungal burden after 3 days of infection. $N=5$; combined data of two independent experiments. Two-tailed Mann–Whitney Test. **k** Serum NGAL and **l** TREM1 of infected mice after 3 days of infection. $N=5$; combined data of two independent experiments. Mann–Whitney Test. **m** Levels of TNFα, IL-1β, and IL-6 in infected kidneys after 3 days of infection. $N=5$; combined data of two independent experiments. Two-tailed Mann–Whitney Test. Results are median ± SEM.

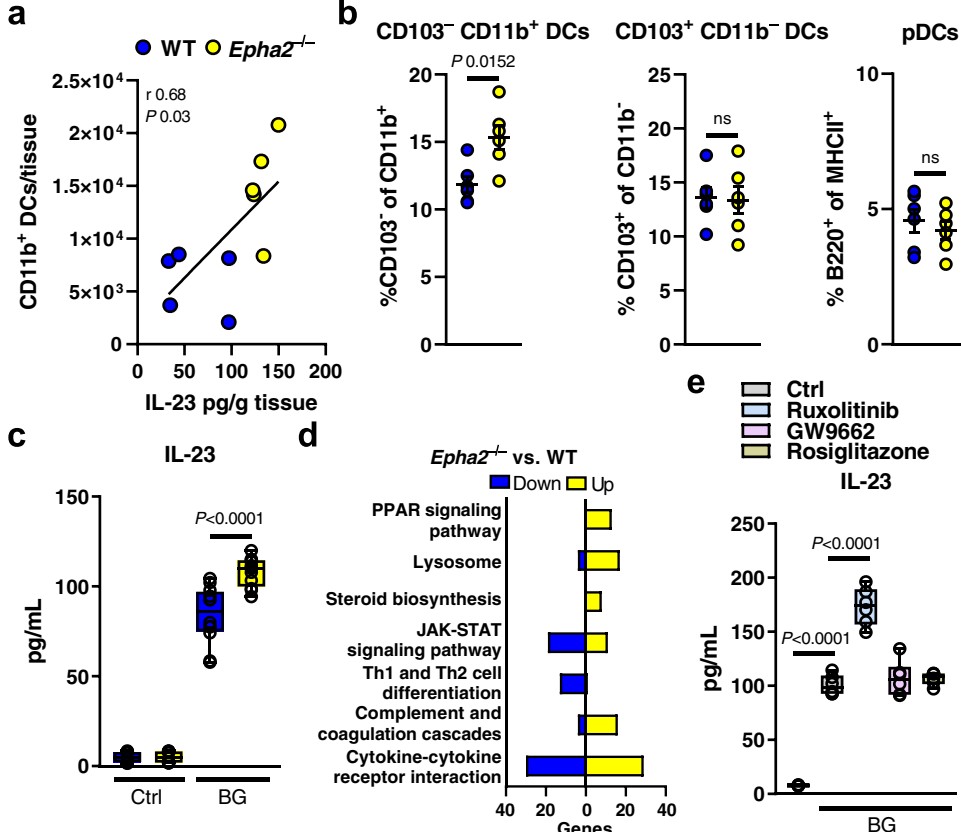

**Fig. 5 | EphA2 signaling impairs IL-23 secretion in BMDCs. a** Correlation of IL-23 kidney levels and cDC infiltration during candidiasis in WT and *Epha2*$^{-/-}$ mice. $N=5$; combined data of two independent experiments. Pearson correlation coefficient. **b** CD103$^-$ CD11b$^+$ DCs, CD103$^+$ CD11b$^-$ DCs, and pDCs in the kidney of wild-type and *Epha2*$^{-/-}$ mice after 3 days of infection. Results are median ($N=6$) ± SEM of two independent experiments combined. ns; No Significance. Two-tailed Mann–Whitney Test. **c** IL-23 levels in supernatants of DCs after 24 h stimulation with β-glucan (curdlan). DCs generated from WT and *Epha2*$^{-/-}$ mice. $N=6$, duplicate. Two-tailed Mann–Whitney Test. Box-and-whisker plots indicating median,

25th/75th percentiles, and the minimum/maximum values. Ctrl, control; BG, β-glucan; Ca, *C. albicans*. **d** Number of up- and down-regulated genes in corresponding KEGG pathways. BMDCs from WT and *Epha2*$^{-/-}$ mice ($N=3$) were stimulated for 6 h with 25 µg/ml curdlan. **e** DC IL-23 secretion treated with indicated inhibitors and stimulated with curdlan. 24 h post stimulation IL-23 levels in supernatants were measured with ELISA. $N=3$; duplicate. Two-tailed Mann–Whitney Test. Box-and-whisker plots indicating median, 25th/75th percentiles, and the minimum/maximum values.

hematopoietic and nonhematopoietic cells contributes to resistance against oral infection. In contrast, both cell compartments promote disease progression during disseminated candidiasis highlighting that the innate immune system responds in an organ-specific manner and varies among infection sites in its ability to control the organism or promote immunopathology. How EphA2 contributes to disease progression in specific host cell types during disseminated

candidiasis is under active investigation. In DCs β-glucan recognition induces several signaling pathways, including AKT, MAPKs, IKK, and NF-κB[102]. Here we show that activation of the β-glucan receptor EphA2 represses IL-23 secretion suggesting that β-glucan recognition stimulates IL-23 expression via Dectin-1/TLR-2[113], but limits the cytokine secretion via non-classical β-glucan recognition. EphA2 activates JAK1/STAT3 signaling[114], and STAT3 deficient DCs exhibit increased

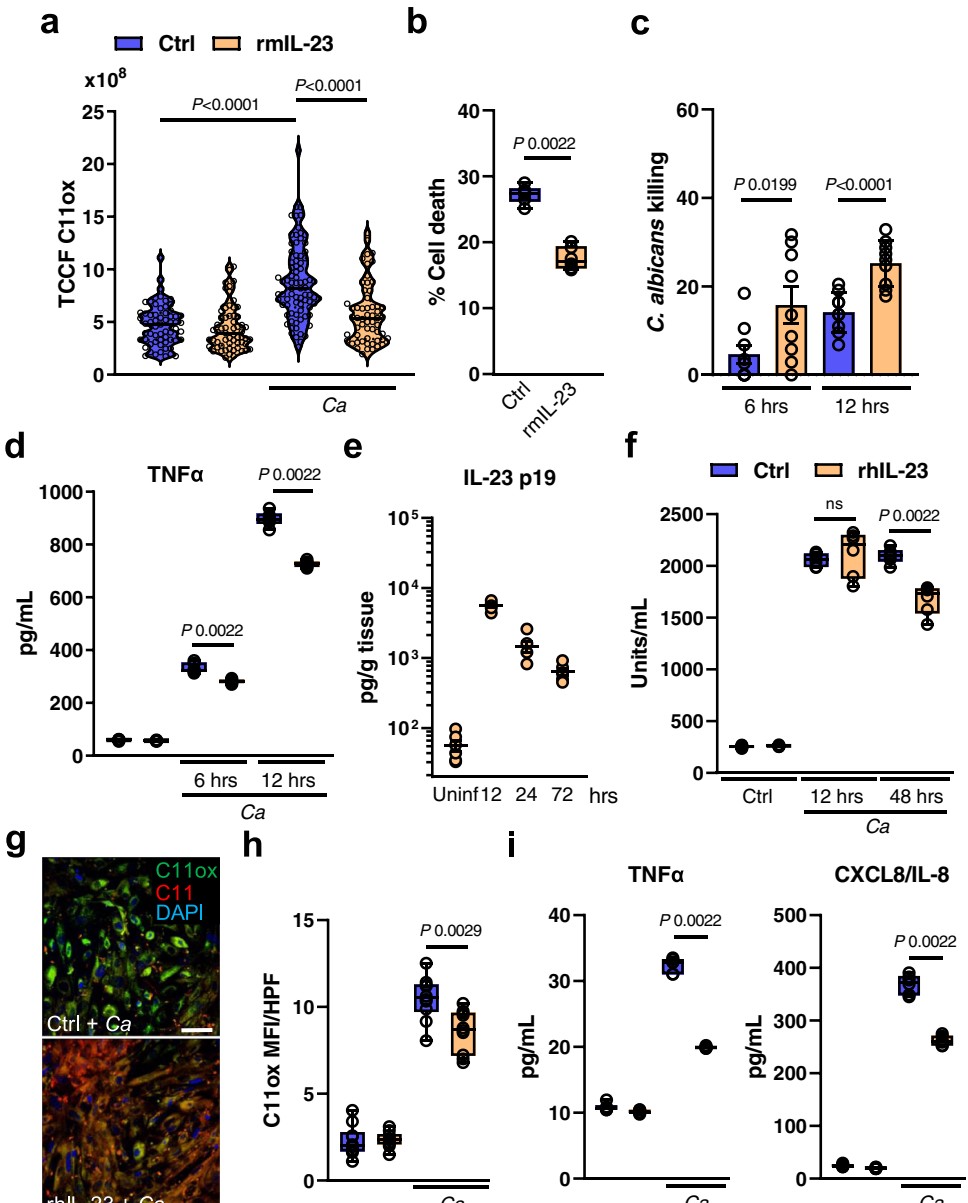

**Fig. 6 | IL-23 prevents ferroptosis in macrophages and renal tubular epithelial cell during *C. albicans* infection. a** BM-derived macrophages were infected with *C. albicans* (MOI 1) for 4 h in the presence and absence of rmIL-23. Total C11ox fluorescence was quantified. Individual cells (*N* = 62 cells for Ctrl; *N* = 60 cells for Ctrl + rmIL-23; *N* = 79 for Ctrl + *Candida*; *N* = 57 for *Candida* + rmIL-23; combined data of three independent experiments) were identified and total fluorescence was measured using ImageJ. Two-tailed Mann–Whitney Test. Violin plot with median and quartiles. **b** BM-derived macrophages were treated with rmIL-23 for 1 h followed by interaction with *C. albicans* (MOI 5) for 4 h. PI⁺ (dead) cells were determined by gating on F4/80⁺ cells. *N* = 3; duplicate. Two-tailed Mann–Whitney Test. Box-and-whisker plots indicating median, 25th/75th percentiles, and the minimum/maximum values. **c** Macrophages-mediated *C. albicans* killing in the presence and absence of rmIL-23. MOI 1. *N* = 3 in triplicate. Bar graphs show mean ± SEM. Two-tailed Mann–Whitney Test. **d** TNFα secretion of BM-derived macrophages during *C. albicans* infection (MOI 1) for indicated time points. *N* = 3, duplicate. Box-and-whisker plots indicating median, 25th/75th percentiles, and the minimum/

maximum values. Two-tailed Mann–Whitney Test. **e** Renal IL-23 p19 levels during disseminated candidiasis. IL-23 p19 levels were determined in kidney homogenates after indicated time points. *N* = 6 for each time point. Mean values ± SEM. **f** LDH release of rhIL-23 (250 µg/ml; 12 and 48 h) treated RTECs followed by infection with *C. albicans* (MOI 5; 5 h). Two-tailed Mann–Whitney Test. *N* = 3 in duplicate. Box-and-whisker plots indicating median, 25th/75th percentiles, and the minimum/maximum values. **g** Representative images of C11 oxidation of RTECs incubated for 48 h with rhIL-23 followed by *C. albicans* infection (MOI 1; 2 h). Scale bar 50 µm. *N* = 10; two independent experiments. **h** Quantification of total C11ox mean fluorescence intensity (MFI) per high power filed (HPF). *N* = 10; two independent experiments. Two-tailed Mann–Whitney Test. Box-and-whisker plots indicating median, 25th/75th percentiles, and the minimum/maximum values **i** CXCL8 and TNFα secretion of RTECs incubated with rhIL-23 during *C. albicans* infection. MOI 5. *N* = 3 in duplicate. Two-tailed Mann–Whitney Test. Box-and-whisker plots indicating median, 25th/75th percentiles, and the minimum/maximum values.

IL-23 production after stimulation[115]. Accordingly, EphA2 deficient DCs and inhibition of JAK signaling in β-glucan stimulated DCs increase IL-23 secretion. Taken together, EphA2-JAK-STAT signaling negatively regulates an inflammatory IL-23 DC phenotype during fungal encounter.

Collectively, our study demonstrates that inflammatory ferrotic host cell death is linked to immunopathology and can be targeted by recombinant cytokine therapy during fungal infection. We postulate that strategies to inhibit ferroptotic cell death during infection will have important therapeutic benefits for fungal diseases.

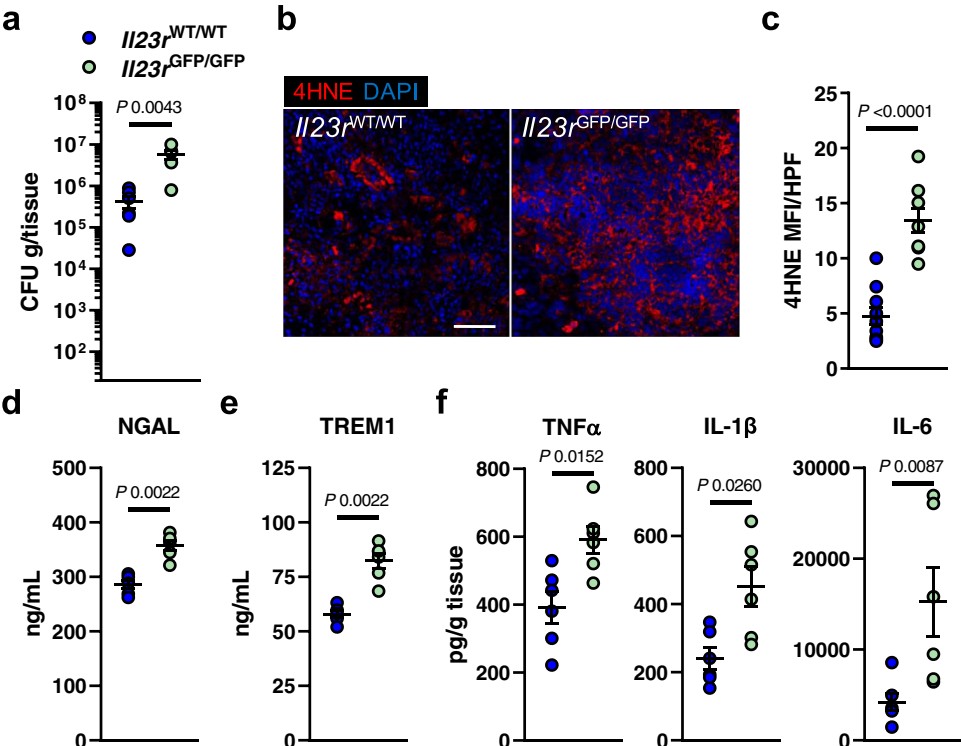

**Fig. 7 | IL-23 receptor signaling decreases ferroptosis and inflammation during disseminated candidiasis. a** Renal fungal burden after 3 days of infection. $N = 6$; combined data of two independent experiments. Two-tailed Mann–Whitney Test. **b, c** Lipid peroxidation (4HNE) in infected kidneys after 3 days of infection. 4HNE shown in red, tissue stained with DAPI (blue). Scale bar 100 μm. Quantification of mean fluorescence intensity (MFI) per high-power field (HPF). $N = 10$ immunofluorescence pictures for $Il23r^{WT/WT}$ and $N = 9$ for $Il23r^{GFP/GFP}$; combined data of two independent experiments. Two-tailed Mann–Whitney Test. **d** Serum NGAL and **e** TREM1 of infected mice after 3 days of infection. $N = 6$; combined data of two independent experiments. Two-tailed Mann–Whitney Test. **f** Levels of TNFα, IL-1β, and IL-6 in infected kidneys after 3 days of infection. $N = 6$; combined data of two independent experiments. Two-tailed Mann–Whitney Test. Results are median ± SEM.

## Methods

### Ethics statement
All animal work was approved by the Institutional Animal Care and Use Committee (IACUC) of the Lundquist Institute at Harbor-UCLA Medical Center.

### Subject details
For in vivo animal studies, age (six-to-ten-week-old) and sex matched mice were used. Animals were bred/housed under pathogen-free conditions at the Lundquist Institute. Mice were housed in groups of five in individually ventilated cages at 22 ± 1 °C, 55 ± 10% relative humidity, 12 h/12 h dark/light cycle, with free access to food and water. Animals were randomly assigned to the different treatment groups. Researchers were not blinded to the experimental groups because the endpoints (survival, fungal burden, cytokine levels) were objective measures of disease severity. $Epha2^{-/-}$ (B6-Epha2$^{tmJrui}$/J) mice were provided by A. Wayne Orr[52]. C57BL/6 control, $Il23r^{WT/GFP}$, and $Clec7a^{-/-}$ mice were purchased from The Jackson Laboratory. $Il23r^{WT/GFP}$ were bred to $Il23r^{WT/GFP}$ to obtain $Il23r^{WT/WT}$ and $Il23r^{GFP/GFP}$ mice. All mice were cohoused for at least 1 week before the experiments.

### Mouse model of HDC
Resistance to disseminated candidiasis was tested in the mouse model of HDC using 6- to 8-week-old mice (C57BL/6 J background) as previously described[20]. The C. albicans SC5314 strain was serially passaged 3 times in YPD broth, grown at 30 °C at 200 rpm for 16–24 h at each passage. Yeast cells were washed, and $2.5 \times 10^5$ or $1.25 \times 10^5$ C. albicans cells injected intravenously via the lateral tail vein. For survival experiments, mice were monitored three times daily and moribund mice were humanely euthanized. To determine organ fungal burden,

mice were sacrificed after 12 h and 4 days of infection, after which the kidneys were harvested, weighed, homogenized, and quantified on Sabouraud dextrose agar plates containing 80 mg/l chloramphenicol. For histology, mouse kidneys were fixed in 10% buffered formalin and embedded in paraffin.

To inhibit ferroptosis during infection mice were treated daily intraperitoneally (start 6 h post infection) with 10 mg/kg Ferrostatin-1 (SelleckChem; >99% purity; S7243 Batch 1) dissolved in 0.9% NaCl. In another experiment mice were treated intravenously 2 days post infection with 12.5 μg/kg recombinant murine IL-23 (R&D Systems; 1887-ML-010/CF; DFXU0821072) for 3 consecutive days. To deplete IL-23 p19 mice were treated intraperitoneally with 250 μg of anti-mouse IL-23 p19 (BioXCell; G23-8) or isotype control (BioXCell; 2A3) in InVivoPure buffer (BioXcell) 6 h post infection, day 2, 4, 6, and 8 post infection. For survival experiments, mice were monitored three times daily and moribund mice were humanely euthanized. For fungal burden and immune responses, mice were euthanized 3 days post infection.

### Mouse model of AKI
Mice were treated intraperitoneally with 750 mg/kg autoclaved zymosan (Sigma-Aldrich) and monitored three times daily and moribund mice were humanely euthanized.

### Immunohistochemistry
Apoptotic cell death was determined as previously described[54]. Briefly, terminal deoxynucleotidyl transferase nick end labeling (TUNEL) staining was performed using the in situ apoptosis detection kit (ApopTag, S7100, Chemicon) according to the manufacturer's protocol with minor modifications. The paraffin-embedded renal sections

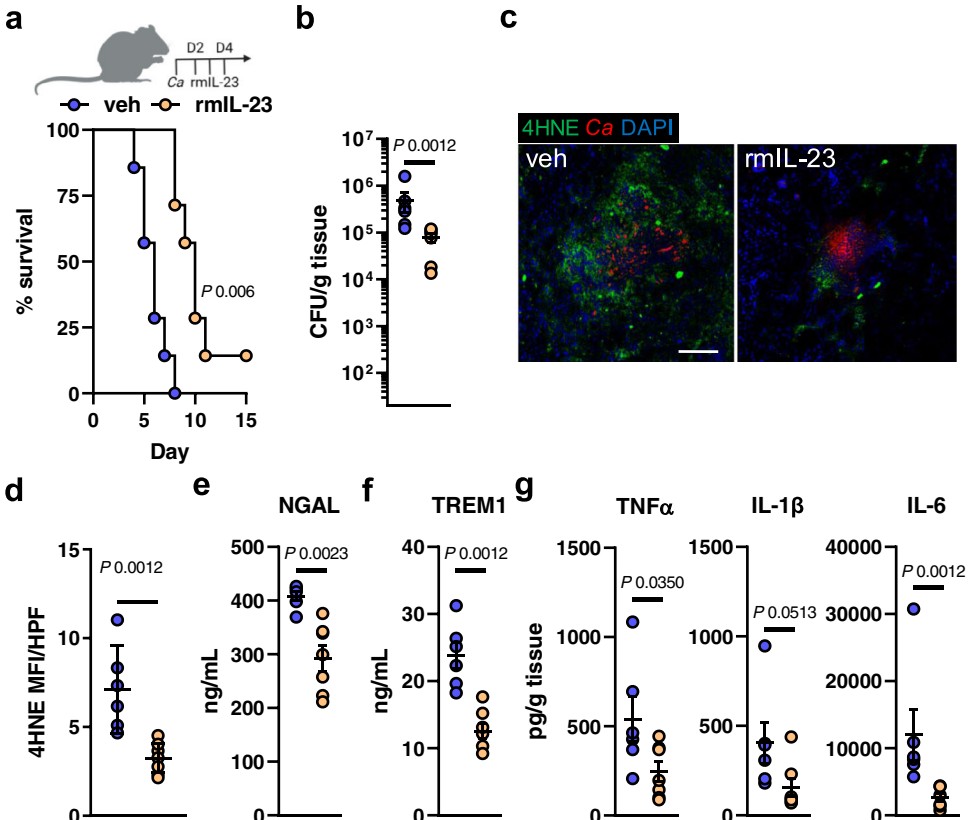

**Fig. 8 | Recombinant IL-23 treatment reduces ferroptosis and improves disease outcomes during *C. albicans* infection. a** (Top) IL-23 treatment scheme during disseminated candidiasis. Created with BioRender.com. (Bottom) Survival of wild-type mice infected intravenously with $2.5 \times 10^5$ SC5314 *C. albicans*. $N = 7$; combined data of two independent experiments. Mice were treated with recombinant murine IL-23 (rmIL-23) or PBS. Mantel–Cox Log-Rank test. **b** Renal fungal burden after 3 days of infection. Single rmIL-23 treatment at day 2. $N = 6$ for vehicle-treated mice and $N = 7$ for rmIL-23-teated mice; combined data of two independent experiments. Mice were treated with recombinant murine IL-23 (rmIL-23) or vehicle (veh; PBS). Two-tailed Mann–Whitney Test. **c**, **d** Lipid peroxidation in infected kidneys after 3 days of infection using 4HNE. Single rmIL-23 treatment at day 2 relative to infection. 4HNE shown in green, *C. albicans* (*Ca*) in red. Tissue is visualized using DAPI. Scale bar 100 μm. Quantification of mean fluorescence intensity (MFI) per high power field (HPF). $N = 6$ for vehicle-treated mice and $N = 7$ for rmIL-23-teated mice; combined data of two independent experiments. Two-tailed Mann–Whitney Test. **e** Serum NGAL and **f** TREM1 of infected mice after 3 days of infection. $N = 6$ for vehicle-treated mice and $N = 7$ for rmIL-23-teated mice; combined data of two independent experiments. Two-tailed Mann–Whitney Test. **g** Levels of TNFα, IL-1β, and IL-6 in infected kidneys after 3 days of infection. $N = 6$ for vehicle-treated mice and $N = 7$ for rmIL-23-teated mice; combined data of two independent experiments. Two-tailed Mann–Whitney Test. Results are median ± SEM.

were placed on poly-L-lysine coated glass slides, deparaffinized in xylene and rehydrated in a graded series of alcohol. Then treated with protease K (20 g/ml) for 15 min at room temperature. Sections were incubated with reaction buffer containing terminal deoxynucleotidyl-transferase at 37 °C for 1 h. After washing with stop/wash buffer, sections were treated with anti-digoxigenin conjugate for 30 min at room temperature and subsequently developed color in peroxidase substrate. The nuclei were counterstained with 0.5% methyl green. TUNEL-positive cells/areas were determined by bright field microscopy. For quantification, apoptotic areas were quantified using PROGRES GRYPHAX® software (Jenoptik).

To determine GPX4 and 4HNE expression, kidneys of WT and *Epha2*$^{-/-}$ mice were harvested 3 days post infection, and snap frozen in Tissue-Tek® OCT. Ten μm kidney sections were fixed with cold acetone, rehydrated in PBS, blocked with BSA, and stained overnight using anti-GPX4 or anti-4HNE (ab125066 and ab46545, respectively). Sections were washed and incubated with anti-rabbit IgG (H + L) coupled with Alexa Fluor™ 488 (Thermo Fisher Scientific). *C. albicans* was detected with an anti-*Candida* antiserum (Biodesign International) conjugated with Alexa Fluor 568 (Thermo Fisher Scientific). To visualize nuclei, cells were stained with DAPI (Prolong Gold® antifade reagent with DAPI, Invitrogen). Images of the sections (*z*-stack) were acquired with a Leica TCS SP8 confocal microscope. To enable comparison of the fluorescence intensities among slides, the

same image acquisition settings were used for each experiment. The relative fluorescence of GPX4 and 4HNE was assessed as mean fluorescence intensity (MFI) per high-power field (HPF) using ImageJ software (v1.8).

### Determination of serum NGAL and TREM1
Serum NGAL and TREM1 were measured at day 3 post-infection. Blood was collected by cardiac puncture from each mouse at the time of euthanasia and stored at −80 °C until use. NGAL and TREM1 concentrations were determined using DuoSet ELISA Kit (DY1857-05 & DY1187, R&D Systems).

### Cytokine and chemokine measurements in vivo
To determine the whole kidney cytokine and chemokine protein concentrations, the mice were intravenously infected with *C. albicans* SC5314. The mice were sacrificed after 3 days post-infection, and their kidney were harvested, weighed, and homogenized. The homogenates were cleared by centrifugation and the concentration of inflammatory mediators was measured using the Luminex multiplex bead assay (Invitrogen).

### Generation of BM chimeric mice
Bone marrow chimeric mice were generated as previously described[40]. Briefly, for BM cell transfers, femurs and tibias from 6- to 8-week-old

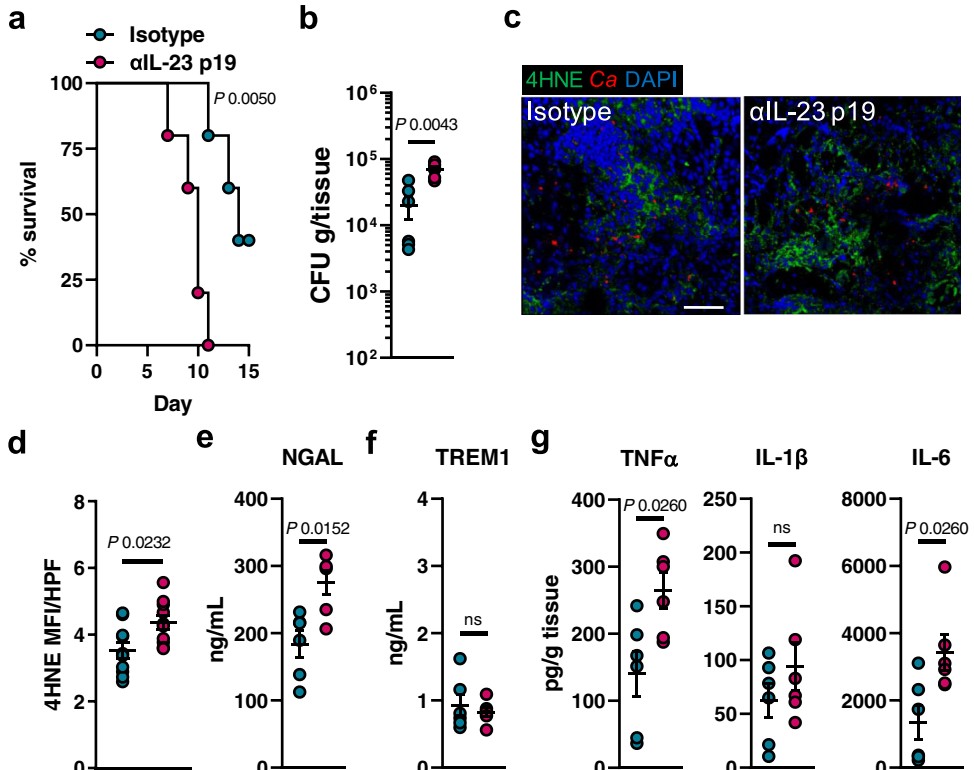

**Fig. 9 | IL-23 depletion in *Epha2*⁻/⁻ mice increases ferroptosis and worsens disease severity. a** Survival of wild-type mice infected intravenously with $2.5 \times 10^5$ SC5314 *C. albicans*. $N = 5$; combined data of two independent experiments. Mantel–Cox Log-Rank test. **b** Renal fungal burden after 3 days of infection. $N = 6$ for each group; combined data of two independent experiments. Two-tailed Mann–Whitney Test. **c, d** Lipid peroxidation in infected kidneys after 3 days of infection using 4HNE. 4HNE shown in green, *C. albicans* (*Ca*) in red. Tissue is

visualized using DAPI. Scale bar 100 μm. Quantification of mean fluorescence intensity (MFI) per high power field (HPF). $N = 10$ immunofluorescence pictures. Two-tailed Mann–Whitney Test. **e** Serum NGAL and **f** TREM1 of infected mice after 3 days of infection. $N = 6$; combined data of two independent experiments. Two-tailed Mann–Whitney Test. **g** Levels of TNFα, IL-1β, and IL-6 in infected kidneys after 3 days of infection. $N = 6$; combined data of two independent experiments. Two-tailed Mann–Whitney Test. Results are median ± SEM.

donor wild-type (*Epha2*⁺/⁺; CD45.1, or CD45.2) and *Epha2*⁻/⁻ (CD45.2) mice were removed aseptically and BM was flushed using cold PBS supplemented with 2 mM EDTA. Recipient wild-type (CD45.1; B6.SJL-*Ptprc^a Pepc^b*/BoyJ) and *Epha2*⁻/⁻ mice were irradiated with 10 Gy and were reconstituted 6 h after irradiation with $2.5 \times 10^6$ *Epha2*⁺/⁺ CD45.2 (WT→WT), *Epha2*⁺/⁺ CD45.1 (wild-type→*Epha2*⁻/⁻), or *Epha2*⁻/⁻ CD45.2 BM cells by lateral tail-vein injection. Mice were given enrofloxacin (Victor Medical) in the drinking water for the first 4 weeks of reconstitution before being switched to antibiotic-free water. Chimeras were infected with *C. albicans* 10 weeks after transplantation. Prior to infection, we confirmed that mice reconstituted with congenic BM stem cells had achieved a satisfactory level of chimerism by assessing the number of CD45.1 and CD45.2 leukocytes in the blood, using flow cytometry (Fig. S1).

### Flow cytometry of infiltrating leukocytes

Immune cells in the mouse kidney were characterized as described. Briefly, mice were infected with *C. albicans* strain SC5314. After 3 days of infection, mice were anesthetized using ketamine/xylazine and perfused with 10 ml of PBS before kidney harvesting. Single-cell suspensions from kidneys were prepared using previously described methods[116]. In brief, kidneys were finely minced and digested at 37 °C in digestion solution (RPMI 1640 with 20 mM HEPES [Gibco] without serum) containing liberase TL (Roche) and grade II DNAse I (Roche) for 20 min with shaking. Digested tissue was passed through a 70-μm filter and washed. The remaining red blood cells were lysed with ACK lysis buffer (Lonza). The cells were suspended in 40% Percoll (GE Healthcare). The suspension was overlaid on 70% Percoll and centrifuged at $836 \times g$ for 30 min at room temperature. The leukocytes

and nonhematopoietic cells at the interphase were isolated, washed 3 times in FACS buffer (0.5% BSA and 0.01% NaN3 in PBS). After washing with FACS buffer, the cell suspension was stained with a Fixable Viability Stain 510 (BD Biosciences), washed, and resuspended in FACS buffer. The single-cell suspensions were then incubated with rat anti-mouse CD16/32 (2.4G2; BD Biosciences) for 10 min (1:100) in FACS buffer at 4 °C to block Fc receptors. For staining of surface antigens, cells were incubated with fluorochrome-conjugated (BUV395, BUV496, BV421, BV711, FITC, PE, PE-Cy7, APC, APC-Cy7) antibodies against mouse CD45 (30-F11; BD Biosciences), Ly6C (AL-21; BD Biosciences), Ly6G (1A8, BioLegend), CD11b (M1/70; BioLegend and BD Biosciences), CD11c (N41, BioLegend), MHCII (M5/114.15.2, BioLegend), and CD206 (C068C2; BioLegend), B220 (RA3-6B2, BioLegend), CD103 (2E7, BD Biosciences), mIL-23R Fab16861, R&D Systems), F4/80 (BM8, BioLegend). All antibodies used for flow cytometry were used 1:100 diluted. The stained cells were analyzed on a FACSymphony system (BD Biosciences), and the data were analyzed using FACS Diva (BD Biosciences) and FlowJo software (Treestar). Only single cells were analyzed, and total cell numbers were quantified using PE-conjugated fluorescent counting beads (Spherotech).

### RNA sequencing

Total RNA was isolated as described before[117]. Briefly, kidneys from infected mice were harvested at 3 days post infection and placed for 1 h in RNAlater solution (Invitrogen). Kidneys were homogenized in Lysing Matrix C tubes (MPBio), following RNA extraction with RNeasy (Qiagen). For RNA sequencing of BMDCs, cells were purified using negative magnetic bead selection (MojoSort Mouse Pan Dendritic Cell Isolation Kit, BioLegend). In all, $2.5 \times 10^6$ BMDCs per well were

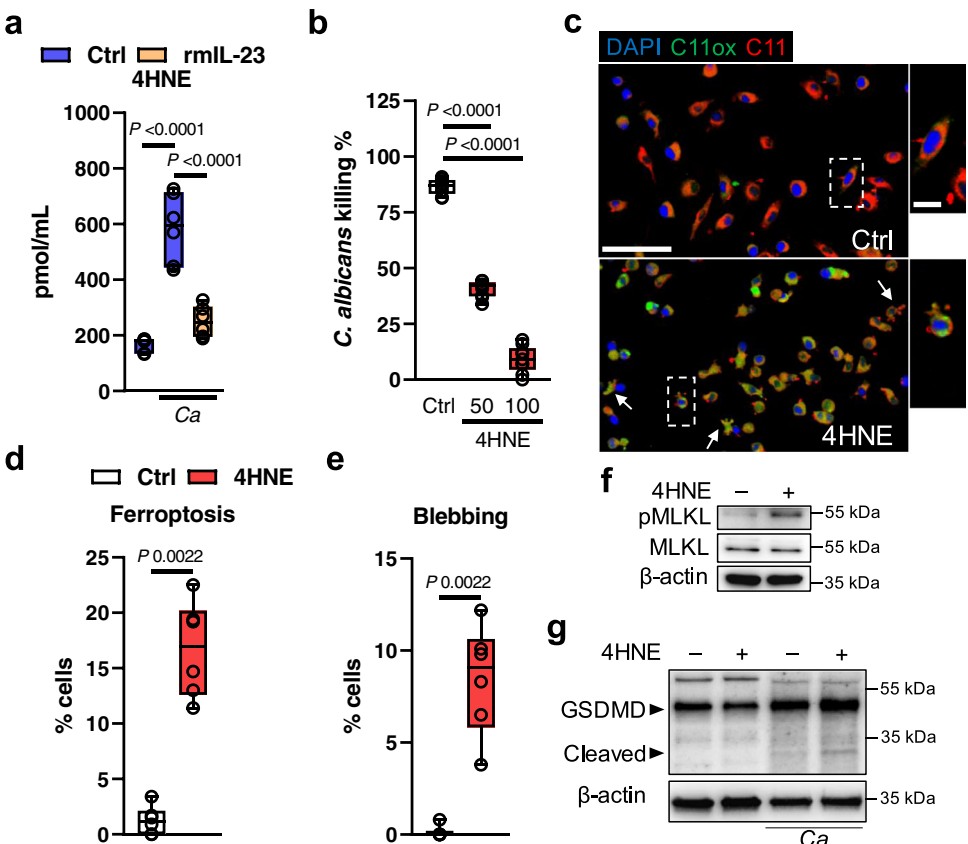

**Fig. 10 | Exogenous 4HNE inhibits macrophage-mediated killing and induces host cell death. a** 4HNE release of BMDMs during *C. albicans* infection. Twelve hours post infection. $N = 3$ in duplicate. Two-tailed Mann–Whitney Test. Box-and-whisker plots indicating median, 25th/75th percentiles, and the minimum/maximum values. **b** Macrophage-mediated killing of cells incubated with 4HNE. MOI 1:5. 12 h post infection. $N = 3$ in triplicate. Ordinary one-way ANOVA corrected for multiple comparison. Box-and-whisker plots indicating median, 25th/75th percentiles, and the minimum/maximum values. **c** Representative images of C11 oxidation of BM-derived macrophages incubated for 4 h with 50 μM 4HNE. Blebbing (apoptotic cells) indicated with arrows. Scale bar 50 (left) and 10 μm (right). Dotted squares indicate magnified areas on the right. $N = 3$ in triplicate. Quantification of **d** ferroptotic (C11ox) and **e** apoptotic (blebbing) macrophages. $N = 3$ in triplicate. Two-tailed Mann–Whitney test. Box-and-whisker plots indicating median, 25th/ 75th percentiles, and the minimum/maximum values. **f** Representative immunoblot of MLKL phosphorylation. BM-macrophages were treated with 50 μM 4HNE for 4 h. Cell lysates were separated via SDS-PAGE and probed for pMLKL, total MLKL, and β-actin. $N = 4$ independent experiments. **g** Representative immunoblot of gasdermin-D cleavage. BM-macrophages were infected with *C. albicans* (MOI 10) for 2 h followed by incubation with 50 μM 4HNE for 4 h. Cell lysates were separated via SDS-PAGE and probed for GSDMD, and β-actin. $N = 4$ independent experiments.

cultured for 6 h in 6-well plates in RPMI complete (R10; 10% HI-FBS, 2 mM L-glutamine, 100 U/ml Penicillin, 100 μg/ml Streptomycin, 50 μM β-ME) and 20 mg/ml GM-CSF in the presence of 25 μg/ml of Curdlan. RNA was isolated using the RNeasy Kit (Qiagen). RNA sequencing was performed by Novogene Corporation Inc. (Sacramento, USA). mRNA was purified from total RNA using poly-T oligo-attached magnetic beads. To generate the cDNA library, the first cDNA strand was synthesized using random hexamer primer and M-MuLV Reverse Transcriptase (RNase H⁻). Second-strand cDNA synthesis was subsequently performed using DNA Polymerase I and RNase H. Double-stranded cDNA was purified using AMPure XP beads and remaining overhangs of the purified double-stranded cDNA were converted into blunt ends via exonuclease/polymerase. After 3' end adenylation a NEBNext Adaptor with hairpin loop structure was ligated to prepare for hybridization. In order to select cDNA fragments of 150–200 bp in length, the library fragments were purified with the AMPure XP system (Beckman Coulter, Beverly, USA). Finally, PCR amplification was performed, and PCR products were purified using AMPure XP beads. The samples were read on an Illumina NovaSeq 6000 with ≥20 million read pair per sample.

### Downstream data processing
Downstream analysis was performed using a combination of programs including STAR, HTseq, and Cufflink. Alignments were parsed using Tophat and differential expressions were determined through DESeq2. KEGG enrichment was implemented by the ClusterProfiler. Gene fusion and difference of alternative splicing event were detected by Star-fusion and rMATS. The reference genome of *Mus musculus* (GRCm38/mm10) and gene model annotation files were downloaded from NCBI/UCSC/Ensembl. Indexes of the reference genome was built using STAR and paired-end clean reads were aligned to the reference genome using STAR (v2.5). HTSeq v0.6.1 was used to count the read numbers mapped of each gene. The FPKM of each gene was calculated based on the length of the gene and reads count mapped to it. FPKM, Reads Per Kilobase of exon model per Million mapped reads, considers the effect of sequencing depth and gene length for the reads count at the same time. Differential expression analysis was performed using the DESeq2 R package (2_1.6.3). The resulting *P*-values were adjusted using the Benjamini and Hochberg's approach for controlling the False Discovery Rate (FDR). Genes with an adjusted *P* value <0.05 found by DESeq2 were assigned as differentially expressed. To allow for log adjustment, genes with 0 FPKM are assigned a value of 0.001. Correlation was determined using the cor.test function in R with options set alternative = "greater" and method = "Spearman." To identify the correlation between the differences, we clustered different samples using expression level FPKM to see the correlation using hierarchical clustering distance method with the function of

heatmap, SOM (Self-organization mapping) and kmeans using silhouette coefficient to adapt the optimal classification with default parameter in R. We used clusterProfiler R package to test the statistical enrichment of differential expression genes in KEGG pathways. The high-throughput sequencing data from this study have been deposited with links to BioProject accession number PRJNA773053 and PRJNA773073 in the NCBI BioProject database.

### RNAscope

*SCL7a11, Gpx4 and Lyz2* mRNA was detected using RNAscope® in situ hybridization. Fresh-frozen sections were thawed and fixed for 15 min in 4% paraformaldehyde (4 C°), and washed in PBS at room temperature. In situ hybridization was performed using the RNAscope® Multiplex v2 Fluorescent Assay (Advanced Cell Diagnostics, Inc.) in strict accordance with the manufacturer's instruction. Probes used were against mouse *Slc7a11* (RNAscope® Probe-Mm-C1, Mm-Slc7a11), *Gpx4* (RNAscope® Probe-Mm-C2, Mm-Gpx4-O1 targeting 12-877 of NM_008162.4) and Lyz2 (RNAscope® Probe-Mm-C3, Mm-Lyz2). The commercially available negative control probe was used, which is designed to target the *DapB* gene from *Bacillus subtilis*. In brief, endogenous peroxidases present in the tissue were blocked with an RNAscope® hydrogen peroxidase solution. Tissue was washed in distilled water, then immersed in 100% ethanol, air dried and a hydrophobic barrier was applied to the slides. The sections were permeabilized with RNAscope® protease III for 30 min at 40 °C. Sections were hybridized with the *Slc7a11, Gpx4, and Lyz2* probes at 40 °C for 2 h. This was followed by a series of amplification incubation steps: Amp 1, 30 min at 40 °C; Amp 2, 30 min at 40 °C; Amp 3, 15 min at 40 °C. Sections were washed with provided washing buffer two times 2 min in between each amplification step. Assignment of Akoya 520 (FITC Slc7a11), 570 (Cy3 Gpx4) and 690 (Cy5 Lyz2) occurred with a peroxidase blocking step sequentially. Finally, DAPI stain was applied, and sections were coverslipped with Invitrogen Prolong Gold® antifade mounting medium. Images were taken with the Leica 3D culture Thunder imaging system. For analysis of two representative sections of the renal cortex were taken for each section for two sections (total of $n = 4$ per animal). Images were analyzed using Image J. Each channel was thresholded with the Otsu filter circularity set to 0.25–1 and particle size set to 10–Infinity. Thresholded images were then analyzed, and total counts were used to represent each gene. Each gene was then normalized to total cell count as assessed by DAPI-positive cell count.

### Bone marrow isolation and cell differentiation

Bone marrow cells were flushed from femurs and tibias using RPMI 1640 medium (Gibco) supplemented with 10% HI-FBS and passed through a 70 μm cell strainer. Bone marrow derived macrophages were generated by growing freshly isolated bone marrow cells from WT and *Epha2*$^{-/-}$ mice in presence of 25 ng/ml M-CSF during 7 days in DMEM supplemented with 10% HI-FBS and Pen/Strep (100U/ml and 100 μ/ml respectively, Gemini Bioproducts) at 37 °C in a humidified atmosphere containing 5% CO$_2$. Bone marrow derived dendritic cells were generated by growing freshly isolated bone marrow cells from WT and *Epha2*$^{-/-}$ mice in presence of 20 ng/ml GM-CSF during 8 days in RPMI complete (R10) at 37 °C in a humidified atmosphere containing 5% CO$_2$. BMDC purity was determined by flow cytometry (>95%).

### Neutrophil killing assay

The capacity of neutrophils to kill *C. albicans* hyphae was determined using the alamarBlue (Invitrogen) reduction as a measure of fungal inactivation. Neutrophils from mice were isolated as described above. Neutrophils were incubated in duplicate wells of flat bottom 96-well plates containing hyphae that had been grown for 3 h with or without serum opsonization (2% heat-inactivated mouse serum; Gemini Bio-

Products), at a neutrophil to *C. albicans* ratio of 1:4 at 37 °C. After 2.5 h, the neutrophils were lysed with 0.02% Triton X-100 in water for 5 min, after which the *C. albicans* hyphae were washed twice with PBS and incubated with 1× alamarBlue (Invitrogen) for 18 h at 37 °C. Optical density at a wavelength of 570 and 600 nm was determined. Neutrophil killing capacity was calculated as the amount of alamarBlue reduced by wells containing *C. albicans* hyphae incubated with and without neutrophils.

### Macrophage killing assay

Bone marrow-derived macrophages were generated as described above. Macrophage killing of *C. albicans* yeast was determined by CFU enumeration. Briefly, unopsonized *C. albicans* SC5314 yeast cells were incubated with BMDMs at a 1:1 ratio for 6 and 12 h, respectively. Macrophages were lysed with 0.02% Triton X-100 in ice-cold water for 5 min, diluted and remaining *Candida* cells were quantitatively cultured. To determine the effect of ferroptosis inhibition during killing BMDMs were incubated with 10 μM Fer-1 or 50 ng/ml rmIL-23 for 1 h prior to infection. To determine the effect of 4HNE on macrophage killing, macrophages were preincubated with 50 or 100 μm 4HNE (50 μM, Selleck Chemicals LLC; S9793 Batch 2).

### Quantification of ferroptosis in vitro

After 7 days of differentiation, BMDMs were collected and seeded on fibronectin-coated glass coverslips in the presence or absence of 10 μM Fer-1 or 50 ng/ml rmIL-23, after 1 h BMDMs were infected with *C. albicans* SC5314 at MOI 1. After 210 min BODIPY ™ 581/591 C11 was added at the concentration of 10 μM for 30 min. Cells were fixed using 2% paraformaldehyde diluted in PBS. After washing the coverslip with PBS, cells were mounted on microscopic glass using ProLong Gold® Antifade with DAPI. The total fluorescent integrated density of the reduced form of BODIPY 581/591 C11 was quantified using ImageJ. To measure individual cellular areas and mean fluorescence, an outline was drawn around each cell, along with several adjacent background readings. Total corrected cellular fluorescence (TCCF) was calculated using the following formula: Integrated density − (area of selected cell × mean fluorescence of background readings).

In another experiment, we quantified 4HNE during macrophage–*C. albicans* interactions. After 240 min, cells were fixed, washed, and permeabilized following staining with anti-4HNE (ab46545; Abcam). TCFF of 4HNE was determined as described above.

### Macrophage damage and survival in vitro

In all, $1 × 10^6$ BMDMs seeded in 24-well plates were incubated with *C. albicans* (MOI 1 and 5, respectively) in presence or absence of 10 μM Fer-1 or 50 ng/ml IL-23. After 4 h, BMDMs were harvested and stained with an antibody against F4/80 (BM8, Biolegend) and propidium iodide (BD Biosciences). The stained cells were analyzed on a FAC-Symphony system (BD Biosciences). In some experiments, lactate dehydrogenase (LDH) release was used as an indicator for host cell damage. LDH release in supernatants was assessed by CytoTox 96 Non-Radioactive Cytotoxicity Assay kit (Promega) according to the manufacturer's protocol.

### Renal tubular epithelial damage and lipid peroxidation

Primary renal proximal tubule epithelial cells (PCS-400-010, ATCC) in a 24-well plate were loaded with $^{51}$Cr overnight. The next day, the cells were incubated with 10 μM Fer-1 or diluent, and then infected with *C. albicans* at a multiplicity of infection of 10. After 8 h, the medium above the cells was collected and the epithelial cells were lysed with RadiacWash (Biodex). The amount of $^{51}$Cr released into the medium and remaining in the cells was determined with a gamma counter, and the percentage of $^{51}$Cr released in the infected cells was compared to the release by uninfected epithelial cells.

The experiment was performed 3 times in triplicate. In some experiments, lactate dehydrogenase (LDH) release was used as an indicator for host cell damage. Briefly, renal tubule epithelial cells were grown to confluence in 24-well tissue culture plate and incubated with recombinant human IL-23 (R&D Systems) at a final concentration of 250 ng/ml for 12 or 48 h prior to infection with *C. albicans*. RTECs were then infected with an MOI of 5. After 5 h incubation, the amount of LDH that was released into the culture medium was quantified using the CytoTox 96 Non-Radioactive Cytotoxicity Assay kit on a BioTek Gen5 plate reader. Renal tubular epithelial cells lipid peroxidation was measured as relative C11ox fluorescence (BODIPY™ 581/591) in mean fluorescence intensity (MFI) per high power field (HPF) using the ImageJ software.

To stain for surface IL-23 receptor RTECs were trypsinized, collected, washed in FACS buffer, and stained with anti-IL-23R antibody (EPR22838-4, Abcam; 1:100), followed by Alexa Fluor 647 Donkey anti-rabbit IgG (Poly4064, BioLegend; 1:100). The stained RTECs were analyzed on a FACSymphony system (BD Biosciences).

### Cytokine measurement in vitro
BMDMs were stimulated for 6 and 12 h with *C. albicans* (MOI 1). RTECs were simulated with *C. albicans* for 12 h (MOI 5). The effect of Fer-1 and RSL3 on cytokine secretion, BMDMs were incubated with Fer-1 (10 μM; S7243 Batch 1, Selleck Chemicals LLC), or RSL3 (0.5 and 1 μM; S8155 Batch 3; Selleck Chemicals LLC). BMDCs were stimulated with curdlan (50 μg/ml; Invivogen) or LPS (1 μg/ml, Sigma Aldrich) for 24 h. Supernatant were collected and cytokines were measured with Luminex Bead array (R&D Systems) or ELISA (TNFα #DY410, and IL-23 #D2300B; R&D Systems). To determine the effect of JAK and PPAR signaling in DCs, BMDCs were incubated with Ruxolitinib (1 μM; S1378 Batch 1; Selleck Chemicals LLC), Rosiglitazone (10 μM; S2556 Batch 2; Selleck Chemicals LLC), and GW9662 (10 μM; S2915 Batch 1; Selleck Chemicals LLC) for 1 h prior stimulation.

### 4HNE release in vitro
In all, $1 \times 10^6$ BMDMs were infected with *C. albicans* (MOI 1) for 12 h in the presence or absence of rmIL-23. 4HNE levels in supernatants were measured via mouse 4-Hydroxynonenal (HNE) ELISA Kit (Biomatik).

### Immunoblotting
In all, $1 \times 10^6$ BMDMs were either treated with 4HNE for 4 h or stimulated for 2 h with *C. albicans* yeast (MOI 10) followed by 4 h treatment with 4 HNE. The supernatant was discarded, and cells were incubated in 55 μl SDS-PAGE buffer. Samples were boiled at 95 °C for 5 min. The lysates were separated by SDS/PAGE, and the proteins were detected by immunoblotting with specific antibodies, including anti-pMLKL (S345) (D6E3G, Cell signaling; 1:1000), anti-MLKL (D6W1K, Cell signaling; 1:1000), anti-cleaved N-terminal GSDMD antibody (EPR20829-408, ab215203, Abcam; 1:1000), anti-rabbit IgG HRP (#7074, Cell Signaling; 1:10,000). To reprobe immunoblots, the membranes were stripped. The blots were developed using enhanced chemiluminescence and imaged with a C400 digital imager (Azure Biosystems). Uncropped raw immunoblots are presented in Fig. S20.

### Quantification and statistical analysis
At least three biological replicates were performed for all in vitro experiments unless otherwise indicated. Data were compared by Mann–Whitney corrected for multiple comparisons using GraphPad Prism (V. 9) software. $P$ values <0.05 were considered statistically significant.

### Reporting summary
Further information on research design is available in the Nature Research Reporting Summary linked to this article.

## Data availability
The authors declare that the data supporting the findings of this study are available within the paper and the accompanying supplementary information files. The high-throughput sequencing data from this study have been deposited with links to BioProject accession numbers PRJNA773053 and PRJNA773073 in the NCBI BioProject database. Source data are provided with this paper.

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

## Acknowledgements

M.S. is supported by NIH grant R00DE026856, R01DE031382, an American Association of Immunologists Careers in Immunology Fellowship, and a Lundquist Seed grant. M.S.L. is supported by the Division of Intramural Research of the NIAID. R.T.W. is supported by NIH grant R15AI133415, and NJ by the Francis family foundation and by NIH National Center for Advancing Translational Science (NCATS) UCLA CTSI Grant Number KL2TR001882, NIH grant R21AI159221, and Grants T32IP4707 and T32KT4708 of the Regents of the University of California Tobacco-Related Diseases Research Program. N.M. and D.A. are supported by a California Institute for Regenerative Medicine Stem Cell Biology Training Grant EDUC4-12837. The content is solely the responsibility of the authors and does not necessarily represent the official views of the National Institutes of Health. We are grateful to A.W. Orr for providing the *Epha2*$^{-/-}$ mice and thank members of the Division of Infectious Diseases at Harbor-UCLA Medical Center for critical suggestions.

## Author contributions

N.M. and M.S. designed the experiments. N.M., N.V.S., D.A., N.J., and M.S. performed the experiments. N.M., N.V.S., D.A., N.J., and M.S. analyzed the data. M.S.L. and R.T.W. provided methodology. M.S. wrote the manuscript. All authors reviewed and edited.

## Competing interests

The authors declare no competing interests.
