## [Peer Review File · Nature Communications]

IL-23 signaling prevents ferroptosis-driven renal immunopathology during candidiasisEditorial Note: Parts of this Peer Review File have been redacted as indicated to maintain the confidentiality of unpublished data.

REVIEWER COMMENTS

Reviewer #1 (Remarks to the Author):

The manuscript by Millet et al. present interesting data showing that animals deficient for the ephrin type-A 2 (EphA2, b-glucan receptor) have increased survival during disseminated candidiasis, less pro-inflammatory cytokines but more IL-23 production and reduced ferroptosis in the kidney. The authors provide evidence in in vitro experiments using BMDM and RTECs showing that *C.albicans* increase cell death, which is prevented by a ferroptosis inhibitor. This ferroptosis inhibitor increases mice survival in disseminated candidiasis model. Furthermore, the authors show that EphA2 null BMDC produce more IL-23 in vitro, and exogenous IL-23 protects against macrophage cell death in vitro. Finally, the administration of rIL-23 in mice during *C.albicans* infection increases survival and reduces ferroptosis. With these data, the authors conclude that: i) *C.albicans* infection causes ferroptosis in the kidney, leading to renal immunopathology that contribute to animal death, ii) Eph2A null animals are partially protected against the renal immunopathology, iii) EphA2 expression in DC limits IL-23 production in vitro and iv) rIL-23 reduces ferroptosis and increases animal survival. Most of the conclusions are strongly supported by experimental data, but others need more experimental support. Specially, the link between EphA2/IL-23/ferroptosis is currently based on indirect correlations and in vitro experiments, and evidences for a direct link must be provided to strengthen the conclusions and to increase the impact of the publications. Below there are some questions/suggestions to improve the quality of this work.

Comment 1

As mentioned above, a direct link between EphA2/IL-23/ferroptosis in vivo must be provided to support conclusions. Currently, this conclusion is supported by indirect observations (EphA2^{-/-} mice have more IL-23 in kidney, EphA2^{-/-} DC in vitro secrete more IL-23, and both EphA2^{-/-} mice and IL-23 administration reduces ferroptosis and increases survival). However, there are some effects seen in vitro with IL-23 that are not recapitulated in vivo in EphA2^{-/-} mice: IL-23 increases macrophage survival and increases *C.albicans* killing (fig. 6B and 6C), while EphA2^{-/-} mice have normal numbers of macrophages (Fig 2C) and equal *C.albicans* load (Figs. 1E,F). Here are some suggestions to provide a direct EphA2/IL-23/ferroptosis link in vivo:

- In the infection model in EphA2^{-/-} animals, treat with neutralizing antibodies against IL-23. If the mechanism is the proposed one, the reduced ferroptosis and increased survival should be lost. A genetic alternative, if available, will be to cross the EphA2^{-/-} with IL-23^{-/-} or IL-23R^{-/-} animals and perform the infection/survival experiment. This experimental setting will provide a strong direct link between EphA2/IL-23 and functional consequences.

- Determine if IL-23 production is truly coming from ex vivo isolated DCs from the kidneys of infected EphA2^{-/-} mice. If the authors can combine this with the staining showed in Fig. S4, they can also determine if the source of IL-23 are conventional or plasmacitoid DCs. These experiments will strengthen the data provided in in vitro differentiated BMDC (Fig. 5B). Additionally, they will discriminate if increased IL-23 is due to increased numbers of DCs (Fig. 2E-F), or due to increased production from DCs (or both).

- Provide evidence of IL-23R expression in kidney macrophages. IL-23R expression is mostly restricted to Th17, Tgd17 and NKT cells, and only minor populations of dendritic cells and macrophages express it. If the mechanism in vivo is mediated by IL-23 effect on macrophages, kidney macrophages should express the IL-23R.

Comment 2

In Fig. 3A-B, only genes for ferroptosis are shown. What about other apoptotic mechanisms? Are necroptosis or pyroptosis affected in EphA2^{-/-} mice? If only ferroptosis is affected it will be good to show it to increase relevance of the pathway.

Comment 3

Data provided regarding the numbers of macrophages in EphA2^{-/-} kidney are confusing. In Fig.2C, macrophages are identified as CD11b+Ly6C-Ly6G-, and numbers in WT and EphA2^{-/-} mice are equal. In Fig. 3D-E, Lyz2 is used to label macrophages, and here there is a reduction of

macrophages. Can authors clarify this difference? According to the in vitro data, increased IL-23 should increase macrophage survival, but this is not clearly detected in vivo. The number of pictures quantified for Fig, 3C need to be increased, 4 pictures per mouse of a complex organ like the kidney seems a low number to get a complete view of what it is going on in the whole organ. Regarding Fig. 3F, pictures need some quantification. For example, Fig. 1E show that pathogen burden is equal in wt and Epha2^{-/-}, but representative picture in fig 3F show decreases Ca staining. Picture quantification will clarify this point.

Minor comments

- EphA2 is not mentioned in the title, while half of the figures are performed in Epha2^{-/-} mice. The authors need to decide where they want to put the strength of the manuscript, only on IL-23/ferroptosis axis, or in the EphA2/IL-23/ferroptosis, and reorganize figures and title accordingly.
- In the Introduction, I am missing more support with clinical relevance to highlight the relevance of the study. For example, how common is disseminated vs mucosal candidiasis in humans? How common is the renal pathology and associated morbidity in patients?
- In the Introduction, I also miss some information about Eph function, signaling, etc. I also suggest to include at the end of this section the main question that the manuscript addresses and a summary of the major findings of the work.
- Figure legends, please provide exact p-value, and specify if data in figure and pooled data from 2 experiments, or if they are representative of 2 experiments.
- What is the mechanism for the protection observed by EphA2^{-/-} deletion in stromal cells (Fig1D). Is the same? Different? Maybe some comments can be added in the discussion section.

Reviewer #2 (Remarks to the Author):

The current report by Millet et al. describes a role for myeloid cell-specific IL-23 signaling in ameliorating immunopathology during candidiasis. Briefly, the authors showed that β -glucan receptor ephrin type-A 2 receptor (EphA2) in myeloid and renal tubular cells drives excessive inflammation in the kidney resulting in kidney immunopathology. Consequently, mice deficient in EphA2 showed reduced renal inflammation and increased protection against candidiasis. Further analyses revealed that EphA2 limits IL-23 secretion from dendritic cells, which is critical for inhibiting highly inflammatory ferroptotic cell death of kidney infiltrating myeloid cells. Overall, this is an interesting finding and a well written paper that addresses mechanisms of organ damage during fatal systemic infection with *Candida albicans* infection. However, there are several major concerns that require additional clarification:

Major concerns:

1. Although the authors mentioned that EphA2 via inhibiting ferroptosis promotes renal immunopathology and subsequently increased mortality following candidiasis, there is no experiment performed to prove this point. In Fig 4H and I, the authors could have easily measure kidney dysfunction and damage of Fer-1 treated mice to prove the point. Also, is there any reason why novel EphA2 inhibitor ALW-II-41-27 (routinely used in cancer research) has not been used to treat fungal infected mice?
2. The paper showed that EphA2 expression is equally important in myeloid and non-myeloid compartments to promote kidney inflammation and damage during candidiasis. However, the impact of IL-23 on ferroptotic cell death of tubular epithelial cells has been ignored and warrants careful consideration.
3. NGAL as a marker for kidney injury should be shown in the infected kidney in addition to serum NGAL level.
4. It is unclear why kidneys of EphA2 deficient mice showed reduced RNA and protein expression of anti-ferroptotic genes SLC7a11 and GPX4, when these mice exhibited reduced ferroptosis and

renal inflammatory changes? Are these antiapoptotic proteins also expressed in tubular epithelial cells of fungal infected kidneys? It seems to me there is no difference in the expression of GPX4 in non LYZ2- cells.

5. The normalization of SLC7a11 and GPX4 RNAscope data by total DAPI+ nuclei is confusing. Since, EphA2 kidneys showed reduced infiltration of myeloid cells in the kidney, the normalization needs to be done with the number of LYZ2+ cells in the kidney parenchyma.

6. The authors must neutralize IL-23 in EphA2^{-/-} mice and show that the protective phenotype is reversed in candidiasis.

Minor:

1. The link between EphA2 and IL-23 production via JAK/STAT and PPAR γ signaling is weak. The author should provide some insight on how EphA2 may be regulating JAK/STAT and PPAR γ in dendritic cells.

2. Most of the inflammatory cytokine and chemokine genes are measured at day 3 post infection. Since immunopathology is generally expected to occur at a later time point, the authors should consider measuring these inflammatory mediators at later time points also. However, this is a minor concern.

Reviewer #3 (Remarks to the Author):

In the manuscript from Swidergall and colleagues they first examined why Eph2a-deficient mice were more resistant to systemic *C. albicans* challenge and identified a novel role of IL-23 in preventing ferroptosis and renal pathology-induced by *C. albicans*. Interestingly, in Eph2a-deficient mice while most cytokines were decreased after *C. albicans* challenge interesting IL-23, IFN γ , and IL-4 were increased in these mice. The authors go on to show that either blocking ferroptosis (using Fer-1) or providing exogenous IL-23 can increase the survival of wild-type mice. These data are robust and provide sound justification for the majority of the authors conclusions. The one critical experiment that is lacking is are the cDCs the essential cell population for secreting the IL-23 in vivo, these could be done using cellular adoptive transfers and/or mixed bone chimeras using mice deficient in those cDC subpopulations. Overall, this is a very exciting paper that could have a dramatic impact on our understanding of the renal immunopathology associated with *C. albicans* infection.

Specific Points:

1.) Figure 2 - What specific cDC subset(s) is(are) increased in the Eph2a-deficient mice (cDC1 or cDC2)? Also, why are these cDCs increasing in cell numbers is it recruitment or local proliferation of those cells? What happens in the Eph2a-deficient mice in the absence of these cDCs (e.g. are the cDC population essential for IL23 production in the mice)?

2.) Figure 4 - When ferroptosis is blocked with Fer-1 does IL-23 and/or cDC numbers change? Does this also result in increased antifungal leukocyte numbers (e.g. neutrophils and monocytes)

3.) Figure 6 - When rmIL23 is provided does this alter the antifungal leukocyte numbers in the mice?

4.) The bone marrow chimerism experiments in Figure 1D are intriguing given that Eph2a is needed in both the radiosensitive and radioresistant cell populations? What happens to the cytokine and cellular response in these mice?

Point-to-point reply

We would like to thank the reviewers for the critical reading and constructive comments that helped us to improve the manuscript. We have addressed all comments and provide a detailed point-to-point reply to all comments below.

Reviewer #1 (Remarks to the Author):

- A direct link between EphA2/IL-23/ferroptosis in vivo must be provided to support conclusions. Currently, this conclusion is supported by indirect observations (EphA2^{-/-} mice have more IL-23 in kidney, EphA2^{-/-} DC in vitro secrete more IL-23, and both EphA2^{-/-} mice and IL-23 administration reduces ferroptosis and increases survival). However, there are some effects seen in vitro with IL-23 that are not recapitulated in vivo in EphA2^{-/-} mice: IL-23 increases macrophage survival and increases *C.albicans* killing (fig. 6B and 6C), while EphA2^{-/-} mice have normal numbers of macrophages (Fig 2C) and equal *C.albicans* load (Figs. 1E,F).

Reply: The overall inflammatory response in Epha2 KO mice is reduced, thus providing an explanation why there is no increase in overall live macrophages (total cell numbers) in these mice (Fig. 2). However, if gated and plotted as % of CD11b⁺ cells we have an increase of live macrophages in Epha2 KO compared to WT mice. This data supports our hypothesis, that Epha2 KO mice have reduced ferroptosis due to increased IL-23 levels, and consequently macrophage survival. We added this data to the revised manuscript (Fig. 2d; see also below).

d Percent of macrophages of CD11b⁺ cells. N=5, combined data of two independent experiments. ns; No Significance. Mann-Whitney Test

As shown in Fig. 1d EphA2 deficiency in both compartments, hematopoietic and stromal, is required for full tolerance during infection. Previously we have shown that during oral mucosal infection EphA2 in different cell types contribute to resistance during infection (PMID: 31291578). Thus, EphA2 on immune or stromal cell types contribute differently to resolution of infection or inflammation which might influence fungal burden. However, we have previously shown that in this mouse model, fungal burden does not correlate with disease outcomes (PMID: 31401652; 34162849), rather immune reactivity and immunopathology correlates with disease outcomes. Our RNASeq data of infected kidneys provided us an overview why these global KO mice are more tolerant- which included a reduction of ferroptotic gene expression. Given that deficiency in stromal EphA2 had similar survival outcomes than deficiency in the hematopoietic compartment,

and IL-23 is secreted by immune cells we focused on IL-23 and ferroptosis. To strengthen our hypothesis that IL-23 receptor signaling prevents ferroptotic cell death we systemically infected $Il23r^{WT/WT}$ and $Il23r^{GFP/GFP}$ mice (unresponsive to IL-23). $Il23r^{GFP/GFP}$ mice had increased renal fungal burden, as well as excessive ferroptosis indicated by 4HNE staining. Furthermore, IL-23R deficiency increased kidney injury and sepsis, while exaggerated inflammation. We added this data in Fig. 7 (see also below). This data shows that IL-23 receptor signaling prevents inflammatory ferroptosis in host cells to improve disease outcomes during disseminated candidiasis.

Figure 7. IL-23 receptor signaling decreases ferroptosis and inflammation during disseminated candidiasis. a Renal fungal burden after 3 days of infection. N=6; combined data of two independent experiments. Mann-Whitney Test. **b-c** Lipid peroxidation (4HNE) in infected kidneys after 3 days of infection. 4HNE shown in red, tissue stained with DAPI (blue). Scale bar 100 μ m. Quantification of mean fluorescence intensity (MFI) per high power field (HPF). N=9-10. Mann-Whitney Test. **d** Serum NGAL and **e** TREM1 of infected mice after 3 days of infection. N=6; combined data of two independent experiments. Mann-Whitney Test. **f** Levels of $TNF\alpha$, $IL-1\beta$, and IL-6 in infected kidneys after 3 days of infection. N=6; combined data of two independent experiments. Mann-Whitney Test. Results are median \pm SEM.

- In the infection model in $EphA2^{-/-}$ animals, treat with neutralizing antibodies against IL-23. If the mechanism is the proposed one, the reduced ferroptosis and increased survival should be lost. A genetic alternative, if available, will be to cross the $EphA2^{-/-}$ with $IL-23^{-/-}$ or $IL-23R^{-/-}$ animals and perform the infection/survival experiment. This experimental setting will provide a strong direct link between $EphA2/IL-23$ and functional consequences.

Reply: As suggested we depleted IL-23 (IL-23 p19 neutralizing antibody) in $EphA2$ KO mice. We found that depletion of IL-23 p19 decreased survival, increased renal fungal burden and

inflammation, as well as ferroptosis (Fig.9; see also below). Thus, the increased IL-23 levels in *Epha2* KO mice contribute to increased tolerance during disseminated candidiasis.

Figure 9. IL-23 depletion in *Epha2*^{-/-} mice increases ferroptosis and worsens disease severity. **a** Survival of wild-type mice infected intravenously with 2.5×10^5 SC5314 *C. albicans*. N=5; combined data of two independent experiments. Mantel-Cox Log-Rank test. **b** Renal fungal burden after 3 days of infection. N=6-7; combined data of two independent experiments. Mann-Whitney Test. **c-d** Lipid peroxidation in infected kidneys after 3 days of infection using 4HNE. 4HNE shown in green, *C. albicans* (Ca) in red. Tissue is visualized using DAPI. Scale bar 100 μ m. Quantification of mean fluorescence intensity (MFI) per high power field (HPF). N=10. Mann-Whitney Test. **e** Serum NGAL and **f** TREM1 of infected mice after 3 days of infection. N=6; combined data of two independent experiments. Mann-Whitney Test. **g** Levels of TNF α , IL-1 β , and IL-6 in infected kidneys after 3 days of infection. N=6; combined data of two independent experiments. Mann-Whitney Test.

- Determine if IL-23 production is truly coming from ex vivo isolated DCs from the kidneys of infected *Epha2*^{-/-} mice. If the authors can combine this with the staining showed in Fig. S4, they can also determine if the source of IL-23 are conventional or plasmacytoid DCs. These experiments will strengthen the data provided in in vitro differentiated BMDC (Fig. 5B). Additionally, they will discriminate if increased IL-23 is due to increased numbers of DCs (Fig. 2E-F), or due to increased production from DCs (or both).

Reply: As suggested by this reviewer we tried to stain intracellularly for IL-23 expression in specific DC populations during infection. We stimulated kidney single cell suspension ex vivo with PMA and ionomycin (different time points). However, we were unable to detect intracellular IL-23 levels in any cell type. We consulted Dr. LeibundGut-Landmann which described initially that IL-23 signaling is required for immune cell survival during infection (PMID 31887131). She told us that they were unable to detect intracellular IL-23 protein levels as well. Since no IL23 reporter mice are available (to the best of our knowledge), we focused on kidney DC subsets which have been

described to produce IL-23 (ex vivo, via marker sorting followed by RNAseq and qPCR), in particular tissue-migratory CD103⁻ CD11b⁺ and CD103⁺ CD11b⁻ DCs (PMID: 26242598). We found an increase of CD103⁻ CD11b⁺ DCs in infected kidneys of *Epha2* KO compared to WT mice, while no differences were observed in CD103⁺ CD11b⁻ DCs, as well as plasmacytoid DCs (pDCs) populations. Thus, previous reports combined with our new data (Fig. 5b; see also below) and that IL-23 depletion in *Epha2* KO mice increases ferroptosis and immunopathology (see above) suggests that an increase in IL-23 reduces ferroptosis and immunopathology in *Epha2* deficient mice.

b CD103⁻ CD11b⁺ DCs, CD103⁺ CD11b⁻ DCs, and pDCs in the kidney of wild type and *Epha2*^{-/-} mice after 3 days of infection. N=6, combined data of two independent experiments. ns; No Significance. Mann-Whitney Test.

- Provide evidence of IL-23R expression in kidney macrophages. IL-23R expression is mostly restricted to Th17, Tgd17 and NKT cells, and only minor populations of dendritic cells and macrophages express it. If the mechanism in vivo is mediated by IL-23 effect on macrophages, kidney macrophages should express the IL-23R.

Reply: While it has already been shown that kidney macrophages express IL-23R during infection (Nur et al.; PMID: 31887131), we performed the suggested experiment. WT mice were infected with 2.5 x10⁵ *Candida* yeast cells. After 3 days of infection the kidneys were harvested and single cell suspensions were stained for IL-23R expression. The histogram (Fig. S11, see also below) shows IL-23R surface expression of live single CD11b⁺ F4/80⁺ macrophages. This, during disseminated candidiasis kidney macrophages express IL-23R.

Fig. S11. CD11b⁺ F4/80⁺ macrophages express the IL-23 receptor during infection. WT mice were infected with 2.5 x10⁵ *Candida* yeast cells. After 3 days of infection. Single cell suspensions of kidneys were stained for IL-23R

expression. Histogram shows representative IL-23R surface expression of live single CD11b⁺ F4/80⁺ macrophages (B). N=4.

-In Fig. 3A-B, only genes for ferroptosis are shown. What about other apoptotic mechanisms? Are necroptosis or pyroptosis affected in Epha2^{-/-} mice? If only ferroptosis is affected it will be good to show it to increase relevance of the pathway.

Reply: We added data showing that ferroptosis, in particular the ferroptosis product 4HNE, promotes other forms of cell death; thus, reducing antifungal effector functions. During *Candida*-macrophage interactions, 4HNE is released which can be blocked by inhibiting ferroptosis with recombinant IL-23. Exogenous 4HNE decreased macrophage-mediated *Candida* killing, while incubation with 4HNE induced apoptosis, necroptosis, and ferroptosis. Furthermore, 4HNE promoted pyroptosis when macrophages were pre-infected with *Candida*. We added this data to the manuscript (Fig. 10; see also below). In this line, key genes involved in these processes were down-regulated in Epha2 KO mice (Fig. S18, see below), which have reduced ferroptosis and 4HNE in their kidneys (Fig. 3h).

Figure 10. Exogenous 4HNE inhibits macrophage-mediated killing and induces host cell death. **a** 4HNE release of BMDMs during *C. albicans* infection. 12 hours post infection. N=3 in duplicate. Mann-Whitney Test. **b** Macrophage-mediated killing of cells incubated with 4HNE. MOI 1:5. 12 hours post infection. N = 3 in triplicate. ANOVA. **c** Representative images of C11 oxidation of BM-derived macrophages incubated for 4 hours with 50 μ M 4HNE. Blebbing (apoptotic cells) indicated with arrows. Scale bar 50 (left) and 10 μ m (right). Dotted squares indicate magnified areas on the right. Quantification of **d** ferroptotic (C11ox) and **e** apoptotic (blebbing) macrophages. N = 3 in triplicate. Mann-Whitney test. **f** Representative immunoblot of MLKL phosphorylation. BM-macrophages were treated with 50 μ M 4HNE for 4 hours. Cell lysates were separated via SDS-PAGE and probed for pMLKL, total MLKL, and β -actin. **g** Representative immunoblot of gasdermin-D cleavage. BM-macrophages were infected with *C. albicans* (MOI 10) for 2

hours followed by incubation with 50 μ M 4HNE for 4 hours. Cell lysates were separated via SDS-PAGE and probed for GSDMD, and β -actin.

Fig. S18. Heatmap of key genes involved in necroptosis, pyroptosis, and apoptosis. Shown are normalized fold changes (FC). RNASeq was performed on mRNA isolated from kidneys of WT and *Epha2*^{-/-} mice after 3 days of infection. N=3 per mouse strain.

- Data provided regarding the numbers of macrophages in *Epha2*^{-/-} kidney are confusing. In Fig.2C, macrophages are identified as CD11b+Ly6C-Ly6G-, and numbers in WT and *Epha2*^{-/-} mice are equal. In Fig. 3D-E, Lyz2 is used to label macrophages, and here there is a reduction of macrophages. Can authors clarify this difference? According to the in vitro data, increased IL-23 should increase macrophage survival, but this is not clearly detected in vivo.

Reply: Lyz2 is a myeloid cell marker which is expressed by macrophages, monocytes, and neutrophils. Thus, we distinguished between Lyz2+ myeloid cells and non-myeloid cells. Given that *Epha2* KO mice have reduced accumulation of neutrophils and monocytes (Fig. 2C) a reduction in Lyz2 expression via RNAScope reflects our flow cytometry data of infected kidneys. For the macrophage survival please see comment above.

The number of pictures quantified for Fig, 3C need to be increased, 4 pictures per mouse of a complex organ like the kidney seems a low number to get a complete view of what it is going on in the whole organ. Regarding Fig. 3F, pictures need some quantification. For example, Fig. 1E show that pathogen burden is equal in wt and *Epha2*^{-/-}, but representative picture in fig 3F show decreases Ca staining. Picture quantification will clarify this point.

The data provided here is from two independent experiments and 4 sections per animal for a total of n=8. Furthermore, we provided quantification of GPX4 protein levels of kidney sections (in independent experiments), which is significantly downregulated in *Epha2* KO mice (added to Fig. S5, see also below). Thus, the RNAScope experiment was verified by protein staining. Further, we quantified lipid peroxidation (specific for ferroptosis) which corresponded to the GSEA analysis that WT mice have more ferroptosis. To show consistency of the GPX4 and 4HNE staining and *Candida* distribution, we added additional pictures in the Fig. S5. While we agree that some variability in *Candida* staining within tissue sections occurs this method cannot distinguish between live and dead *Candida* cells, while the CFU enumeration in Fig. 1 shows no difference between live organisms.

Fig. S5. GPX4 and 4HNE levels in infected kidneys of Wt and *Epha2*^{-/-} mice. **A** (Left) GPX4 quantification of mean fluorescence intensity (MFI) per high power field (HPF). N=7. Mann-Whitney Test. (Right) GPX4 shown in green, *C. albicans* (Ca) in red. Tissue is visualized using DAPI. **B** 4HNE quantification of mean fluorescence intensity (MFI) per high power field (HPF). N=10-11. Mann-Whitney Test. (Right) 4HNE shown in green, *C. albicans* (Ca) in red. Tissue is visualized using DAPI.

Minor comments

- EphA2 is not mentioned in the title, while half of the figures are performed in *Epha2*^{-/-} mice. The authors need to decide where they want to put the strength of the manuscript, only on IL-23/ferroptosis axis, or in the EphA2/IL-23/ferroptosis, and reorganize figures and title accordingly.

Reply: Given that we added 3 figures solely about IL-23 and/or ferroptosis we prefer to keep title and figure organization as is.

- In the Introduction, I am missing more support with clinical relevance to highlight the relevance of the study. For example, how common is disseminated vs mucosal candidiasis in humans? How common is the renal pathology and associated morbidity in patients?

Reply: We added this information (lines 31-33; 39-42).

- In the Introduction, I also miss some information about Eph function, signaling, etc. I also suggest to include at the end of this section the main question that the manuscript addresses and a summary of the major findings of the work.

Reply: We added a summary paragraph (lines 64-69). The function of EphA2 signaling during fungal infection is stated in lines 59-63.

- Figure legends, please provide exact p-value, and specify if data in figure and pooled data from 2 experiments, or if they are representative of 2 experiments.

Reply: We revised all figures which present now the exact p values, as well as figure legends. All in vivo work is combined data from two independent experiments.

- What is the mechanism for the protection observed by EphA2^{-/-} deletion in stromal cells (Fig1D). Is the same? Different? Maybe some comments can be added in the discussion section.

Reply: This is an excellent point. This observed phenotype is currently under investigation. However, we believe that any analysis of EphA2 in stromal cells is beyond the scope of this manuscript (IL-23 signaling and ferroptosis). We added this to the discussion section (lines 326-334).

Reviewer #2 (Remarks to the Author):

We appreciate the reviewer's comment "interesting finding and a well written paper that addresses mechanisms of organ damage during fatal systemic infection with *Candida albicans* infection"

Major concerns:

1. Although the authors mentioned that EphA2 via inhibiting ferroptosis promotes renal immunopathology and subsequently increased mortality following candidiasis, there is no experiment performed to prove this point. In Fig 4H and I, the authors could have easily measure kidney dysfunction and damage of Fer-1 treated mice to prove the point. Also, is there any reason why novel EphA2 inhibitor ALW-II-41-27 (routinely used in cancer research) has not been used to treat fungal infected mice?

Reply: We added data showing that Fer-1 treatment reduced fungal burden, kidney injury (NGAL) and sepsis (TREM1), as well as inflammation (cytokines TNF α , IL-1 β , and IL-6) (Fig. 4; see also below). This new data shows that *C. albicans* induces ferroptosis to promote inflammation, kidney injury, and consequently disease progression during systemic disease.

j Renal fungal burden after 3 days of infection. N=5; combined data of two independent experiments. Mann-Whitney Test. **k** Serum NGAL and **l** TREM1 of infected mice after 3 days of infection. N=5; combined data of two independent experiments. Mann-Whitney Test. **m** Levels of TNF α , IL-1 β , and IL-6 in infected kidneys after 3 days of infection. N=5; combined data of two independent experiments. Mann-Whitney Test.

This reviewer mentioned the EphA2 inhibitor ALW-II-41-27. However, this inhibitor is not specific for EphA2. It inhibits several other enzymes e.g Src or EphB2. EphB2 together with Dectin-1 recognizes beta-glucan to induce anti-fungal immunity (PMID: 32312989; 33685996). Thus, treating mice with this inhibitor will not specifically inhibit EphA2 during infection. We performed this experiment

[REDACTED]

2. The paper showed that Epha2 expression is equally important in myeloid and non-myeloid compartments to promote kidney inflammation and damage during candidiasis. However, the impact of IL-23 on ferroptotic cell death of tubular epithelial cells has been ignored and warrants careful consideration.

Reply: As suggested we performed experiments in which we analyzed ferroptosis in renal tubular epithelial cells (RTECs). In particular, recently Tsokos and colleagues (PMID: 33956666) found that these stromal cells express the IL-23R. We found that IL-23 treatment increased IL-23R expression in RTECs. Incubation of RTECs with recombinant human IL-23 decreased *Candida*-mediated cell death in a time dependent manner. Furthermore, prolonged exposure of RTECs to IL-23 decreased *Candida*-induced lipid peroxidation and reduced cytokine secretion of RTECs during *Candida* infection. This data suggest that IL-23R signaling inhibits ferroptosis in renal tubular epithelial cells, as well as macrophages. We added this data in Fig. 6 f-i (see also below).

f LDH release of rhIL-23 (250 $\mu\text{g/ml}$) treated RTECs followed by infection with *C. albicans* (MOI 5). Mann-Whitney Test. N=3 in duplicate. **g** Representative images of C11 oxidation of RTECs incubated for 48 hours with rhIL-23 followed by *C. albicans* infection (MOI 1; 2 hours). Scale bar 50 μm . **h** Quantification of total C11ox fluorescence per high power field (HPF). N=10. Mann-Whitney Test. **i** CXCL8 and TNF α secretion of RTECs incubated with rhIL-23 during *C. albicans* infection. MOI 5. Mann-Whitney Test.

3. NGAL as a marker for kidney injury should be shown in the infected kidney in addition to serum NGAL level.

Reply: We measured NGAL levels in kidney homogenates and found that Epha2 KO mice have a reduction in kidney injury as well. We added this data to the manuscript for WT and Epha2 KO mice (Fig. 1h, see also below).

h Serum and kidney NGAL, and **I** TREM1 levels after 3 days of infection. N=6; combined data of two independent experiments. Mann-Whitney Test.

4. It is unclear why kidneys of EphA2 deficient mice showed reduced RNA and protein expression of anti-ferroptotic genes SLC7a11 and GPX4, when these mice exhibited reduced ferroptosis and renal inflammatory changes? Are these antiapoptotic proteins also expressed in tubular epithelial cells of fungal infected kidneys? It seems to me there is no difference in the expression of GPX4 in non LYZ2⁻ cells. 5. The normalization of SLC7a11 and GPX4 RNAscope data by total DAPI⁺ nuclei is confusing. Since, EphA2 kidneys showed reduced infiltration of myeloid cells in the kidney, the normalization needs to be done with the number of LYZ2⁺ cells in the kidney parenchyma.

Reply: Although considered anti-ferroptotic genes our data indicated that these genes are strongly upregulated when Lyz2 positive cells actively undergo ferroptosis. Previous work showed that viral infection upregulates SLC7A11 and GPX4 expression in host cells (PMID: 35105963). Thus our data supports previous findings. Furthermore, these genes are expressed in non-Lyz2 positive cells as indicated in the graph (Fig 3e; see also below). While Slc7a11 and GPX4 is upregulated in Lyz2 positive cells, no differences were observed in Lyz2 negative cells suggesting that the mechanism of ferroptosis and anti-ferroptosis differs depending on the cell type. We are planning to follow up on this in future experiments. As suggested we added quantification of SLC7a11 in Lyz2 positive and negative cells (Fig 3e; see also below), as we have done it for GPX4.

e *SLC7a11*⁺ and f *GPX4*⁺ particles of LY2Z positive and negative cells. N=8.

6. The authors must neutralize IL-23 in *Epha2*^{-/-} mice and show that the protective phenotype is reversed in candidiasis.

Reply: See reply to reviewer 1 in which we show that IL-23 depletion in *Epha2* KO mice increases ferroptosis, injury and inflammation.

Minor:

1. The link between EphA2 and IL-23 production via JAK/STAT and PPAR α signaling is weak. The author should provide some insight on how EphA2 may be regulating JAK/STAT and PPAR α in dendritic cells.

Reply: We have previously shown that EphA2 is required for STAT3 activation (PMID: 29133884), while recently it was shown that *Epha2* signals via JAK1/STAT3 (PMID: 33626345.). This is in our opinion sufficient (together with the provided data) to make the conclusion that EphA2/JAK/STAT signaling represses IL-23 secretion.

2. Most of the inflammatory cytokine and chemokine genes are measured at day 3 post infection. Since immunopathology is generally expected to occur at a later time point, the authors should consider measuring these inflammatory mediators at later time points also. However, this is a minor concern.

Reply: Historically we perform immune response analysis a day prior the first mouse dies during survival experiments. We have done this approach in Fig. 1 and 2. We could run new experiments with a lower inoculum and measure cytokine responses at a later time point. However, in this mouse model, the animals die from sepsis (PMID: 15962230) and immunopathology (PMID: 22916017). Given that these responses are only delayed when mice are infected with a lower inoculum, measuring pro-inflammatory cytokines, injury and sepsis markers in WT and *Epha2* KO mice infected with a lower dose would not add anything new to the manuscript. Furthermore, this would increase the overall mice numbers used in this study.

Reviewer #3 (Remarks to the Author):

We appreciate the reviewer's comment "a very exciting paper that could have a dramatic impact on our understanding of the renal immunopathology associated with *C. albicans* infection"

Specific Points:

1.) Figure 2 - What specific cDC subset(s) is(are) increased in the Eph2a-deficient mice (cDC1 or cDC2)? Also, why are these cDCs increasing in cell numbers is it recruitment or local proliferation of those cells? What happens in the Eph2a-deficient mice in the absence of these cDCs (e.g. are the cDC population essential for IL23 production in the mice)?

Reply: See comment reviewer 1 in which we discuss the increase of CD103⁻ CD11b⁺ DCs in infected kidneys of Eph2 KO compared to WT mice.

Regarding a specific deletion of these cell types. Since no floxed EphA2 mice are available (Wayne Orr, LSU Shreveport, is working to generate these; personal communication) this will be impossible to do at this time.

2.) Figure 4 - When ferroptosis is blocked with Fer-1 does IL-23 and/or cDC numbers change? Does this also result in increased antifungal leukocyte numbers (e.g. neutrophils and monocytes).

Reply: We performed experiments in which we measured cytokine responses (including IL-23). While TNF α , IL-1 β , and IL-6 (see also comment reviewer 2) were significant downregulated in Fer-1-treated mice compared to the vehicle group, IL-23 levels are unaffected during Fer-1 treatment (we added this data to the rebuttal letter; see below) suggesting that IL-23 producing cells including CD11b⁺ DCs are not changing in the absence of ferroptotic cells death.

Fig. R2. IL-23 levels in infected kidneys after 3 days of infection. N=5; combined data of two independent experiments. Mann-Whitney Test. Results are median \pm SEM.

We decided to measure cytokines instead of performing immunophenotyping because mice of these experiments are not perfused so that the kidney homogenates can be used to generate other data such as fungal burden. Also this allowed us to collect blood for serum markers (Fig. 4;

see comment reviewer 2). Furthermore, this will also ensure to keep the mice numbers as low as possible.

3.) Figure 6 - When rmlL23 is provided does this alter the antifungal leukocyte numbers in the mice?

Reply: As stated above we chose to perform additional experiments with homogenates rather than single cell suspensions so that we can measure inflammation, injury, and sepsis. We found that treatment with recombinant IL-23 decreased injury, sepsis and inflammation (Fig. 8e-g, see also below). IL-23 has been shown to induce Th17 cell proliferation. To exclude a contribution of this cell type, we measured the surrogate markers IL-17A and IL-22 and found no differences in these cytokines suggesting that the observed phenotypes are independent of the IL-23/Th17 axis. We added this data set in Fig. S15. Nur et al. (PMID 31887131) showed that deficiency in IL-23 results in a loss of myeloid, but not lymphoid cells, upon *C. albicans* infection. This together with our new data (IL-23R KO mice have increased inflammation, injury and sepsis, as well as host cell ferroptosis) supports our hypothesis that ferroptotic cell death exaggerates inflammation and immunopathology (see also comment reviewer 1).

e Serum NGAL and **f** TREM1 of infected mice after 3 days of infection. N=6-7; combined data of two independent experiments. Mann-Whitney Test. **g** Levels of TNF α , IL-1 β , and IL-6 in infected kidneys after 3 days of infection. N=6-7; combined data of two independent experiments. Mann-Whitney Test.

4.) The bone marrow chimerism experiments in Figure 1D are intriguing given that Eph2a is needed in both the radiosensitive and radioresistant cell populations? What happens to the cytokine and cellular response in these mice?

Reply: Please see comment reviewer 1 in which we want to follow up on EphA2 function in specific cell types in future work.

REVIEWERS' COMMENTS

Reviewer #1 (Remarks to the Author):

The authors have successfully addressed all my comments and concerns, and the new data included have greatly improved the quality and impact of the research. In my opinion, the manuscript is acceptable for publication in Nature Communications

Reviewer #2 (Remarks to the Author):

The authors have adequately addressed all my concerns.

Reviewer #3 (Remarks to the Author):

The authors very thoroughly addressed the significant concerns/weaknesses identified in the original review. As such, this manuscript is significantly improved and ties together a lot of the original loose end. Very nice work!

Point-to-point reply

We would like to thank the three reviewers for the critical reading and constructive comments.

REVIEWERS' COMMENTS

Reviewer #1 (Remarks to the Author):

The authors have successfully addressed all my comments and concerns, and the new data included have greatly improved the quality and impact of the research. In my opinion, the manuscript is acceptable for publication in Nature Communications

Reviewer #2 (Remarks to the Author):

The authors have adequately addressed all my concerns.

Reviewer #3 (Remarks to the Author):

The authors very thoroughly addressed the significant concerns/weaknesses identified in the original review. As such, this manuscript is significantly improved and ties together a lot of the original loose end. Very nice work!